# Sensory neurons couple arousal and foraging decisions in *Caenorhabditis elegans*

Elias Scheer, Cornelia I Bargmann*

Lulu and Anthony Wang Laboratory of Neural Circuits and Behavior, The Rockefeller University, New York, United States

**\*For correspondence:**
cori@rockefeller.edu

**Competing interest:** The authors declare that no competing interests exist.

**Abstract** Foraging animals optimize feeding decisions by adjusting both common and rare behavioral patterns. Here, we characterize the relationship between an animal's arousal state and a rare decision to leave a patch of bacterial food. Using long-term tracking and behavioral state classification, we find that food leaving decisions in *Caenorhabditis elegans* are coupled to arousal states across multiple timescales. Leaving emerges probabilistically over minutes from the high arousal roaming state, but is suppressed during the low arousal dwelling state. Immediately before leaving, animals have a brief acceleration in speed that appears as a characteristic signature of this behavioral motif. Neuromodulatory mutants and optogenetic manipulations that increase roaming have a coupled increase in leaving rates, and similarly acute manipulations that inhibit feeding induce both roaming and leaving. By contrast, inactivating a set of chemosensory neurons that depend on the cGMP-gated transduction channel TAX-4 uncouples roaming and leaving dynamics. In addition, *tax-4*-expressing sensory neurons promote lawn-leaving behaviors that are elicited by feeding inhibition. Our results indicate that sensory neurons responsive to both internal and external cues play an integrative role in arousal and foraging decisions.

## eLife assessment

This is an **important** study on how behavioral context affects decision making in the nematode *C. elegans*. Behavioral analyses at multiple time scales combined with genetic and neuronal manipulations revealed how arousal states affect decision making. The results and interpretations are **convincing**. This work will be of interest to both neuroscientists and ecologists.

## Introduction

Persistent behavioral states subdivide continuous behavior into discrete modules that accomplish adaptive goals (*Tinbergen, 1951*). An example from behavioral ecology is foraging behavior, which is composed of locomotion patterns that unfold across short and long timescales. For example, the brief darting maneuvers that drive prey capture in hunting zebrafish are embedded within persistent locomotory arousal states that last for minutes (*Marques et al., 2020*). In recent years, classical ethological studies of behavioral states have been supplemented with machine vision, clustering, and classification algorithms that are well-suited to quantitative and systematic analysis (*Berman et al., 2014*; *Schwarz et al., 2015*; *Wiltschko et al., 2015*). A current challenge is to identify circuit mechanisms that couple long-term behavioral states to short-term motor actions in foraging and other goal-directed behaviors.

Foraging behaviors and the neural mechanisms that generate them have been a fruitful subject of study in the nematode *Caenorhabditis elegans*. While exploring a bacterial lawn, which corresponds

**eLife digest** When animals forage for food, they show distinct behavioral patterns in their movement. For instance, the nematode worm *Caenorhabditis elegans* shows two long-term behavioral states when exploring a patch of food: dwelling, when it moves slowly in a small area, and roaming, when it makes quick and wide-ranging movements. The worms will also occasionally suddenly decide to leave a piece of food and go explore the rest of their environment.

Scientists know that the likelihood of the worms either roaming or dwelling is regulated by neurons passing molecules, such as serotonin and dopamine, to one another. However, it is not known how these two long-term behavioral states impact the momentary decision to leave a piece of food, and which mechanisms may regulate this coupling.

To investigate, Scheer and Bargmann tracked the movement of genetically modified *C. elegans* and characterized their behavior. This revealed that the decision to leave food is not random but a distinct choice that primarily happens when worms are roaming. A characteristic signature of this response was that worms briefly accelerate immediately before leaving.

Following this discovery, Scheer and Bargmann identified sensory neurons that are involved in this process. As well as detecting external sensory cues, these neurons also integrate internal signals, like whether the animal can eat, to specify how often a worm will leave food.

The implications of this research extend beyond the realm of tiny nematodes. This study provides a new framework to examine the relationship between long-term behavior and momentary decision making. Such insights are crucial in understanding brain function across different organisms, including humans. It paves the way for further research into how behavior is regulated on multiple timescales in the brain.

to a food patch, *C. elegans* adopts two major behavioral states: roaming and dwelling (*Ben Arous et al., 2009*; *Fujiwara et al., 2002*). Roaming is an aroused state characterized by high locomotion speed with few reversals and turns, whereas dwelling is defined by low speed and high reorientation rates (*Ben Arous et al., 2009*; *Flavell et al., 2013*; *Fujiwara et al., 2002*). A third behavioral state, quiescence, is induced by stress, molting, or satiety (*Gallagher et al., 2013*; *Hill et al., 2014*; *Raizen et al., 2008*; *Van Buskirk and Sternberg, 2007*). The bistability of roaming and dwelling states is governed by neuromodulatory signaling via the neuropeptide Pigment Dispersing Factor (PDF-1), which promotes roaming by signaling through its cognate receptor PDFR-1, and the biogenic amine serotonin, which promotes dwelling by activating the serotonin-gated chloride channel MOD-1. These modulators act through distributed circuits within the 302 neurons of the animal's nervous system, with multiple sources of each modulator and multiple receptor-expressing sites (*Flavell et al., 2013*).

Another foraging behavior studied in *C. elegans* is the decision to stay or leave a lawn of bacterial food. Leaving is infrequent on high quality food lawns but increases when feeding is impaired, the bacterial food is inedible, or food is depleted (*Bendesky et al., 2011*; *Milward et al., 2011*; *Olofsson, 2014*; *Shtonda and Avery, 2006*). Animals also leave more frequently from lawns of pathogenic bacteria (*Pujol et al., 2001*) and lawns spiked with aversive or toxic compounds (*Melo and Ruvkun, 2012*; *Pradel et al., 2007*).

The genes that regulate lawn-leaving behavior overlap with those that regulate roaming and dwelling states, although these behaviors have largely been studied separately. For example, roaming, dwelling, and leaving are all strongly influenced by sensory neurons, particularly those that express the cGMP-gated channel *tax-4* (*Davis et al., 2023*; *Fujiwara et al., 2002*; *McCloskey et al., 2017*; *Milward et al., 2011*; *Supplementary file 1*), and by endocrine DAF-7 (TGF-beta) and Insulin/DAF-2 (insulin receptor) signaling pathways whose peptide ligands are produced by sensory neurons (*Ben Arous et al., 2009*; *Bendesky et al., 2011*).

Here, we characterize the behavioral patterns and circuit mechanisms that accompany lawn leaving decisions on short and long timescales. Using high-resolution imaging to monitor the behaviors of many individual animals, we find that leaving is a discrete behavioral motif that is generated probabilistically during the high-arousal roaming state. A signature of leaving behavior is a rapid acceleration in speed immediately before the leaving event. Roaming states evoked by neuromodulatory mutations, acute feeding inhibition, or optogenetic circuit manipulation all stimulate lawn leaving

accompanied by the rapid acceleration motif. In addition, chemosensory neurons play an unexpectedly central role in linking internal states and behavioral dynamics.

## Results

### Animals make foraging decisions at the boundaries of bacterial food lawns

To study foraging decisions at high resolution, we filmed and quantified the behavior of individual animals on small lawns of bacterial food (~3 mm diameter) on an agar surface (*Figure 1A and B*). During a 40-min assay, animals spent ~97% of their time on the lawn, and preferentially occupied the edge of the bacterial lawn where bacterial density was highest (*Figure 1C*, *Figure 1—figure supplement 1A–D*). As a result, the tip of the animal's head encountered the lawn boundary approximately once per minute. A few of these encounters resulted in a lawn leaving event, in which the animal exited the bacterial lawn (*Figure 1D*, *Figure 1—video 1*). Lawn-leaving occurred in 26% of the assays, with a mean value of one event per 95 min (*Figure 1H–I*, *Figure 1—source data 1*).

Once an animal left the lawn, it typically remained outside of the food for 1–2 min before re-entering the lawn (*Figure 1—figure supplement 1E*, *Figure 1—source data 1*). Most edge encounters were not followed by leaving; instead, animals that poked their head outside of the lawn either continued forward locomotion (head poke-forward), paused (head poke-pause), or executed a reversal (head poke-reversal; *Figure 1E–G*, *Figure 1—video 1*). Head poke-reversals were the most common event after lawn edge encounter (1.1 events/min), followed by head poke-forward and head poke-pause events; lawn leaving events were the least frequent (*Figure 1I*, *Figure 1—source data 1*).

High resolution analysis revealed characteristic behavioral dynamics prior to different events. Each head poke encounter with the lawn edge was preceded by an increase in speed over three seconds, and resolved by a reversal, pause, or continued forward movement within 5 s (*Figure 1J*). Lawn leaving events encompassed a longer acceleration that began 30 s before the leaving event (*Figure 1J*). At longer time scales, the average speed over 1 min before and after the edge encounter varied slightly for different classes of events, with head poke-forward events and lawn-leaving associated with the highest speeds. The different behavioral patterns preceding lawn leaving and head pokes suggest that lawn leaving is a distinct foraging decision, and not a random resolution of an edge encounter. In particular, it suggests that lawn leaving is linked to a persistent behavioral state reflected in ongoing locomotion speed.

### Lawn leaving behavior is associated with high arousal states on short and long timescales

To gain further insight into the behavioral states preceding lawn leaving, we coarse-grained our behavioral measurements into 10 s intervals ('bins') and expanded the time axis to allow analysis across different durations. On average, a slow rise in speed began two minutes before lawn leaving, with a rapid acceleration in the last minute before leaving (*Figure 2A*). We analyzed these behaviors in the context of the well-characterized roaming and dwelling states (*Ben Arous et al., 2009*; *Flavell et al., 2013*; *Fujiwara et al., 2002*). Training a two-state Hidden Markov Model (HMM) based on absolute speed and angular speed (reorientations) recovered a bifurcation of roaming and dwelling states (*Figure 2—figure supplement 1A–D*; Methods). Using these criteria, wild type animals on small lawns roamed for a median of 12% of the duration of the assay (*Figure 2—figure supplement 1C*, *Figure 2—source data 1*). Although 80% of lawn leaving events occurred within roaming states, less than 1% of the 10 s roaming bins contained a lawn leaving event (*Figure 2B*). Five minutes before leaving, 34% of animals were roaming, a fraction that steadily increased such that 80% of animals were roaming immediately before leaving (*Figure 2C*, *Figure 2—source data 1*). By contrast, the fraction of animals roaming before head poke-reversals increased only slightly before the event (*Figure 2—figure supplement 3C*). Within roaming states, the average speed was constant until the rapid acceleration ~30 s prior to lawn leaving (*Figure 2D*). To ask if the brief acceleration accounts for the apparent increase in roaming before lawn leaving, we compared the duration of roaming states that preceded lawn leaving to those that did not. We found that roaming states directly before lawn leaving were slightly longer than other roaming state, arguing that acceleration events alone do not drive the correlation between roaming and lawn leaving (*Figure 2—figure supplement 1E*).

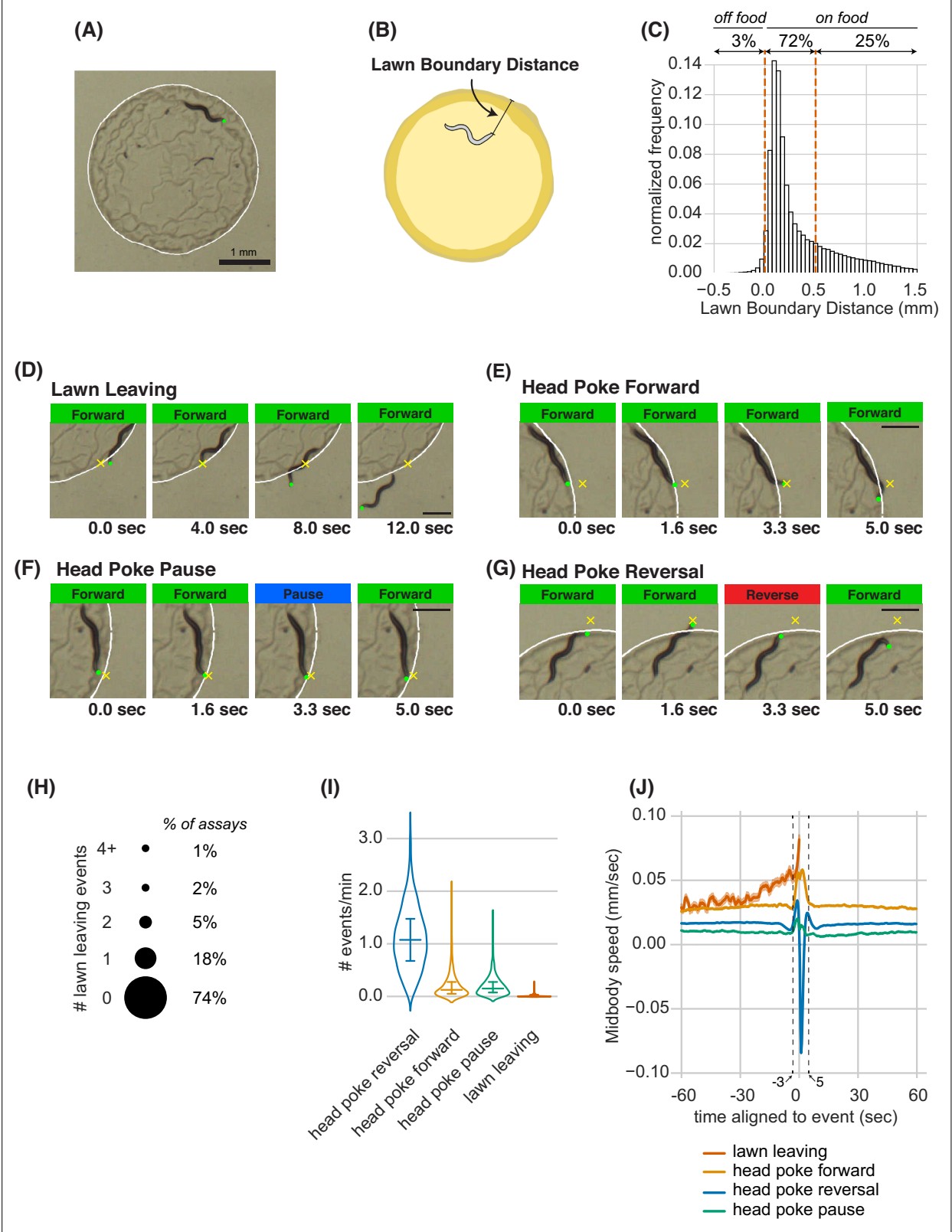

**Figure 1.** Animals make foraging decisions at the boundaries of bacterial food lawns. (**A**) Image of *C. elegans* on a small lawn of bacteria. Head is indicated with a green dot, lawn boundary is indicated with a white line. Scale bar is 1 mm. (**B**) Schematic depicting lawn boundary distance measured as the distance from the head to the closest point on the lawn boundary. (**C**) Empirical distribution of lawn boundary distance in wild type datasets. Positive values indicate distances inside the lawn, negative values indicate distances outside the lawn. (**D–G**) Four example images from behavioral

*Figure 1 continued on next page*

*Figure 1 continued*

sequences generating different types of foraging decisions. Scale bar is 0.5 mm in all panels. Head is indicated with a green dot in all panels. In E-G, yellow X indicates the maximum displacement of the head outside the lawn during head pokes. (**D**) Lawn leaving occurs when an animal approaches lawn boundary and fully crosses through it to explore the bacteria-free agar outside the lawn. Yellow X indicates the position on the lawn boundary encountered by worm's head before exiting the lawn. (**E**) Head Poke Forward occurs when animal continues forward movement on the lawn during and after poking its head outside the lawn. (**F**) Head Poke Pause occurs when animal pauses following head poke before resuming forward movement on the lawn. (**G**) Head Poke Reversal is generated by a reversal following head poke before resuming forward movement on the lawn. (**H**) Number of lawn leaving events per animal in 40-min assay. Bubble sizes are proportional to the percentage of animals that execute each number of lawn leaving events within a 40-min assay. (**I**) Frequency of different foraging decisions. (**J**) Midbody speed aligned to different foraging decision types. Wild type dataset (C,H,I,J): n=1586 animals. Violin plots in (I) show median and interquartile range. In all time-averages, dark line represents the mean and shaded region represents the standard error. See *Figure 1—source data 1*.

The online version of this article includes the following video, source data, and figure supplement(s) for figure 1:

**Source data 1.** Quantification of various lawn boundary interaction behaviors from experiments in Figure 1.

**Figure supplement 1.** Small bacterial lawns are denser at the lawn boundary, and quantification of duration outside the lawn (**A**) A small lawn of bacteria expressing green fluorescent protein (GFP) imaged in the GFP channel.

**Figure 1—video 1.** Lawn boundary encounters lead to different behavioral responses: lawn leaving, head poke forward, head poke pause, and head poke reversal.

https://elifesciences.org/articles/88657/figures#fig1video1

The roaming-dwelling HMM was initially developed on uniform bacterial lawns (*Flavell et al., 2013*), which elicit more roaming than the small lawns used here (*Figure 2—figure supplement 1F–H*, *Figure 2—source data 1*). Expanding this model to include posture information identified eight subclasses of dwelling states (*Cermak et al., 2020*). To explore alternative analysis methods, we generated an HMM behavioral state model trained on our experimental conditions. An Autoregressive Hidden Markov model (AR-HMM) (*Buchanan et al., 2017*; *Linderman et al., 2020*), trained over an expanded set of locomotion parameters, converged on four states that largely reflect forward locomotion speed and related features (*Figure 2—figure supplement 2A–K*, Methods). These four states provided a different segmentation of locomotory arousal that partly overlaps with roaming and dwelling states (*Figure 2—figure supplement 2M–N*). Most lawn leaving events occurred during the high speed, high arousal AR-HMM state 3 (*Figure 2E*). Animals spent a median of 9% of the time in state 3 (*Figure 2—figure supplement 2K*, *Figure 2—source data 1*), but 5 min before leaving, 26% of animals were in state 3, which ramped to 82% immediately before leaving (*Figure 2F*, *Figure 2—source data 1*). Like roaming, state 3 increased only slightly before head poke reversals (*Figure 2—figure supplement 3E*). Animals across all behavioral states – roaming, dwelling, and the four AR-HMM states – were similarly distributed across the bacterial lawn, with strong enrichment near the lawn boundary (*Figure 2—figure supplement 4A–D*).

Both the two-state model and the four-state model show that arousal states increase before lawn leaving over at least two phases: an enrichment of a high arousal state 3–5 min before leaving, followed by a rapid acceleration in the last 30 s. Because the widely used two-state roaming-dwelling model captured the key features of lawn-leaving as well as these alternative methods, we applied that analysis to subsequent experiments.

## Food intake regulates arousal states and lawn leaving

Animals roam more and leave lawns more frequently when bacteria are hard to ingest or have low nutritional value (*Ben Arous et al., 2009*; *Shtonda and Avery, 2006*). Conversely, simply increasing food concentration suppressed roaming and leaving on small lawns (*Figure 3A–B*, *Figure 3—source data 1*; *Bendesky et al., 2011*). To separate the behavioral effects deriving from sensation of food odors from ingestion of food, we treated *E. coli* OP50 bacteria with aztreonam, an antibiotic that renders bacteria inedible to *C. elegans* by preventing bacterial cell septation (*Ben Arous et al., 2009*; *Gruninger et al., 2008*). Both lawn leaving and roaming were dramatically increased on aztreonam-treated *E. coli* lawns (*Figure 3C–D*, *Figure 3—figure supplement 1A–B*, *Figure 3—source data 1*, *Supplementary file 2*). Animals on aztreonam-treated lawns maintained the characteristic speed acceleration in the last 30 s before leaving (*Figure 3—figure supplement 1C*).

Locomotion speed is regulated by direct interoception of pharyngeal pumping, the motor behavior associated with intake of bacterial food (*Rhoades et al., 2019*). To ascertain whether similar mechanisms

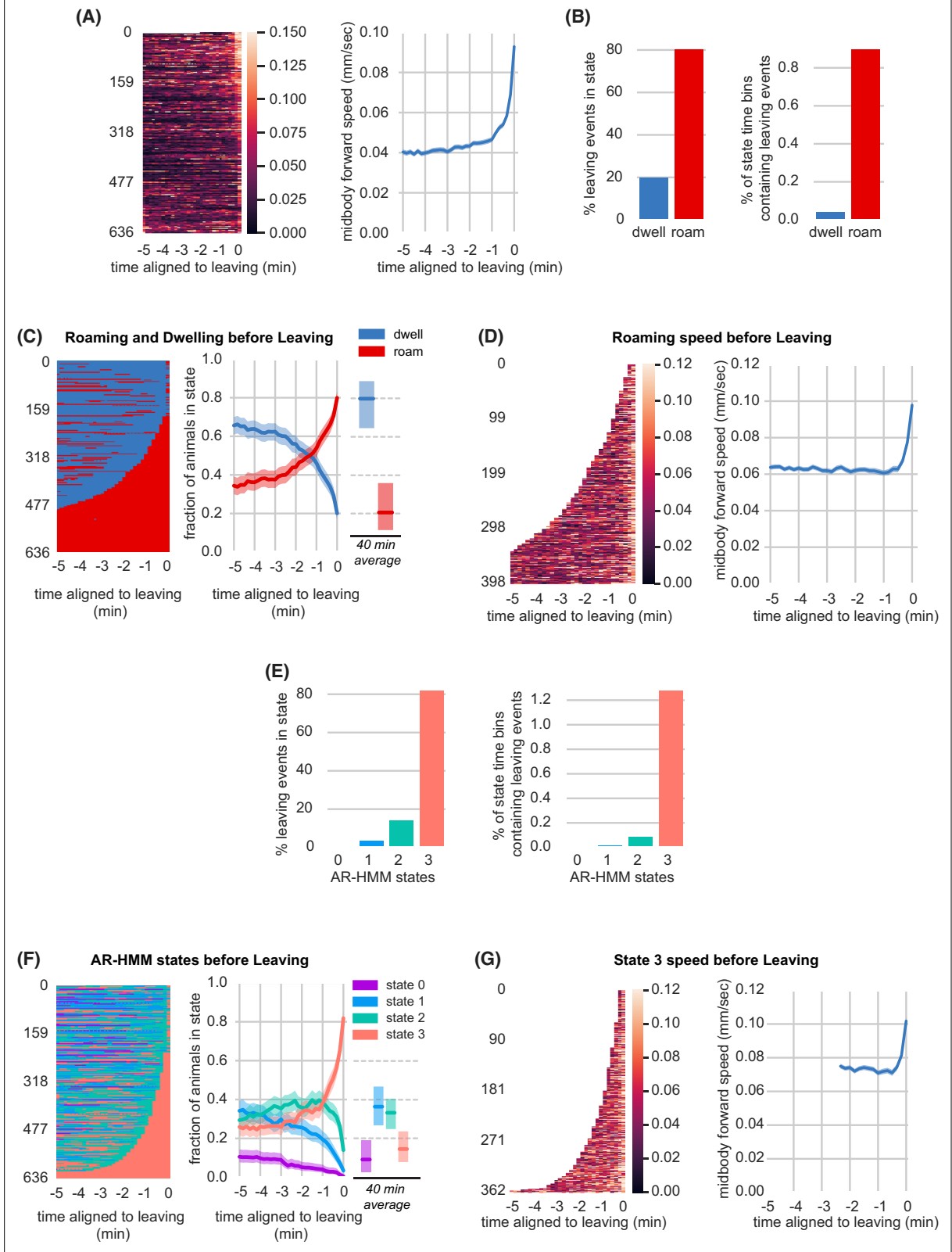

**Figure 2.** Lawn leaving is associated with high arousal states on multiple timescales. (**A**) Midbody forward speed aligned to lawn leaving. Left, heatmap of individual speed traces. Right, mean midbody forward speed computed across the heatmap traces. White space indicates missing data or times when animal was off the lawn. (**B**) Overlap of lawn leaving events with roaming and dwelling. Left, percentage of leaving events found in each state. Right, percentage of 10 s roaming or dwelling bins that contain a leaving event. (**C**) Roaming and dwelling aligned to lawn leaving. Left, heatmap of roaming/

*Figure 2 continued on next page*

*Figure 2 continued*

dwelling state classifications aligned to lawn leaving event. Center, fraction of animals in roaming or dwelling state prior to leaving. Right, total fraction of time spent roaming and dwelling in all assays that included a lawn-leaving event (n=371). Median is highlighted and interquartile range is indicated by shaded area. (**D**) Roaming speed aligned to lawn leaving. Left, heatmap showing speed of roaming animals before lawn leaving. Right, mean roaming speed computed at times when less than 10% of aligned traces had missing data. White space indicates times when animals were not roaming before leaving. (**E**) Overlap of lawn leaving events with AR-HMM states. Left, percentage of leaving events found in each state. Right, percentage of 10 s bins per AR-HMM state that contain a leaving event. (**F**) AR-HMM states aligned to lawn leaving. Left, heatmap of AR-HMM state classifications. Center, fraction of animals in each state prior to leaving. Right, total fraction of time spent in each AR-HMM state in all assays that included a lawn-leaving event (n=371). Median is highlighted and interquartile range is indicated by shaded area. (**G**) State 3 speed aligned to lawn leaving. Left, heatmap showing speed of animals in state 3 before lawn leaving. Right, mean state 3 speed computed at times when less than 10% of aligned traces had missing data. White space indicates times when animals were not in state 3 before leaving. See *Figure 2—source data 1*.

The online version of this article includes the following video, source data, and figure supplement(s) for figure 2:

**Source data 1.** Quantification of behavioral states surrounding lawn leaving events from experiments in *Figure 2*.

**Figure supplement 1.** Roaming and dwelling states on small bacterial lawns.

**Figure supplement 2.** Modeling behavioral states across locomotory feature dimensions using an Autoregressive Hidden Markov Model.

**Figure supplement 3.** Head Poke Reversals are associated with small changes in arousal states.

**Figure supplement 4.** Lawn boundary distance distributions by HMM state (**A–D**) Empirical distributions of lawn boundary distance by HMM state.

**Figure 2—video 1.** Example of animal behavior with roaming and dwelling annotated.

https://elifesciences.org/articles/88657/figures#fig2video1

**Figure 2—video 2.** The same animal and time steps as in *Figure 2—video 1*, except annotated with AR-HMM states instead of roaming and dwelling.

https://elifesciences.org/articles/88657/figures#fig2video2

affect arousal and lawn leaving, we acutely inhibited pharyngeal pumping by optogenetic activation of the red-shifted channelrhodopsin ReaChR in pharyngeal muscle (*Lin et al., 2013*; *Figure 3E–J*). Acute feeding inhibition strongly stimulated both roaming and lawn leaving (*Figure 3E–H*, *Figure 3—figure supplement 1D–E*, *Figure 3—source data 1*, *Supplementary file 2*).

While roaming and head pokes began immediately upon feeding inhibition, leaving events did not occur immediately but instead accumulated throughout the 10-min stimulation interval (*Figure 3I*). In each case, leaving was preceded by a 30 s acceleration in speed (*Figure 3J*). Thus, feeding inhibition elicits both roaming and lawn-leaving behaviors, and lawn-leaving occurs probabilistically during roaming states.

## Parallel regulation of arousal states and leaving by neuromodulatory mutants

To further examine the relationship between arousal states and lawn leaving, we examined mutants with known alterations in roaming and dwelling. Animals deficient in serotonin (*tph-1*), dopamine (*cat-2*), or the neuropeptide receptor NPR-1 (*npr-1*) roam at a higher rate than wild type (*Cermak et al., 2020*; *Cheung et al., 2005*; *Flavell et al., 2013*; *Omura et al., 2012*; *Sawin et al., 2000*). We found that each of these mutants showed increased lawn leaving compared to wild-type controls, strengthening the observed correlation between roaming and leaving rates (*Figure 4A–C and E–G*, *Supplementary file 2*). In addition, the fraction of animals roaming increased over several minutes before leaving events, as was observed in wild type animals (*Figure 4I–K*). Finally, each mutant accelerated during the 30 s prior to leaving (*Figure 4M–O*). Thus, although the molecular basis of arousal is different in each of these mutants, the overall dynamics of roaming and lawn leaving are preserved across genotypes.

Animals lacking the G-protein-coupled receptor Pigment Dispersing Factor Receptor (PDFR-1) roam less than wild type (*Flavell et al., 2013*; *Ji et al., 2021*; *Meelkop et al., 2012*). On the small lawns used here, *pdfr-1* mutants roamed less, but left food lawns at the same low rate as wild type (*Figure 4D and H*). Roaming rates increased similarly during the 3 min prior to lawn leaving in wild type and *pdfr-1* animals, suggesting that the coupling of roaming and leaving does not require PDFR-1 signaling (*Figure 4L*). Although their basal locomotion speed is lower (*Flavell et al., 2013*; *Ji et al., 2021*), *pdfr-1* did accelerate slightly before lawn leaving (*Figure 4P*). In summary, neuromodulatory mutants varied in the fraction of time spent roaming and dwelling, but in each case lawn-leaving behaviors were coupled to roaming and a brief speed acceleration.

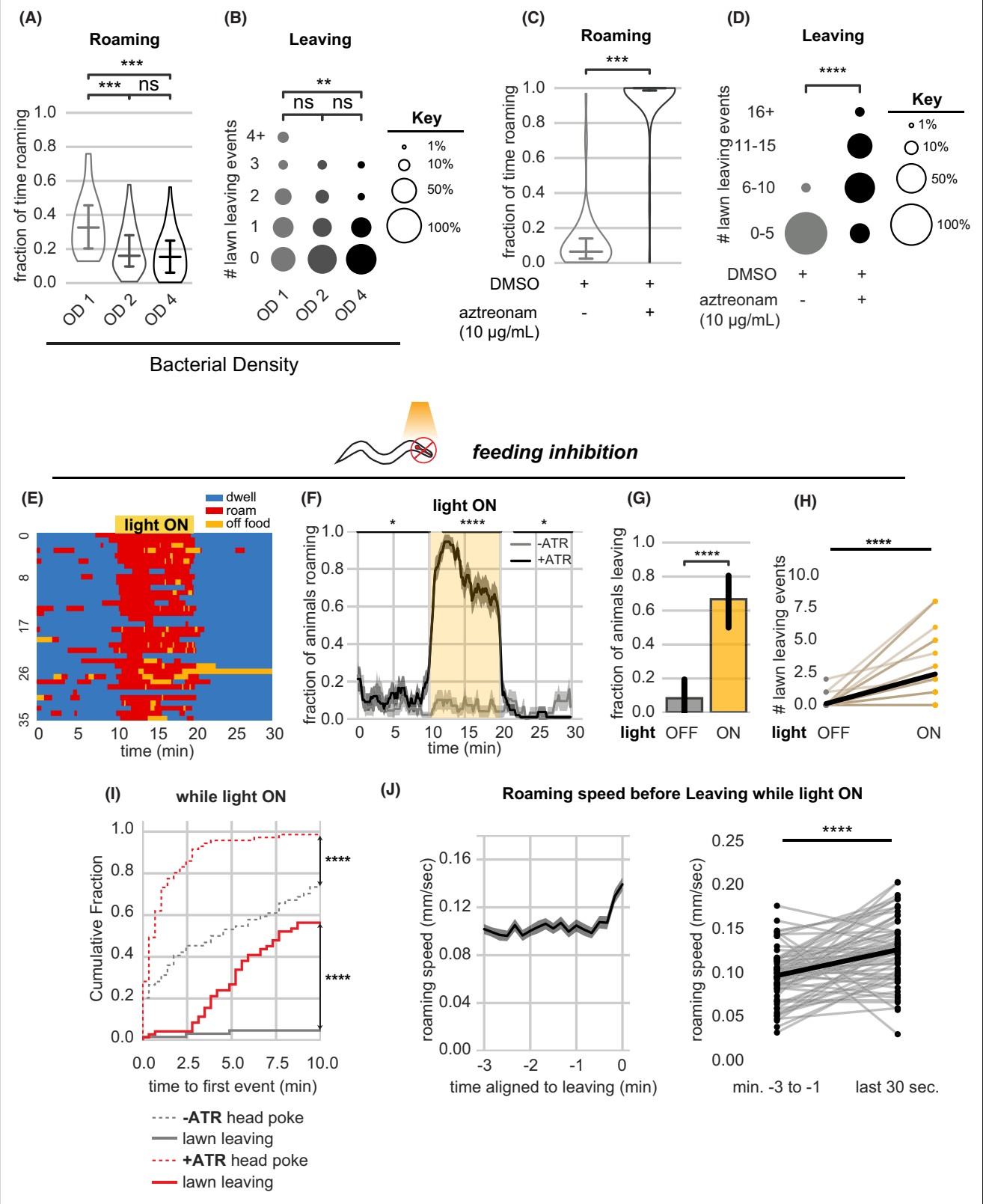

**Figure 3.** Food intake regulates arousal states and lawn leaving. (**A–B**) Increasing bacterial density suppresses roaming and leaving. (**A**) Fraction of time roaming on small lawns seeded with bacteria of different optical density (OD). OD1: n=45, OD2: n=46, OD4: n=47, Statistics by one-way ANOVA with Tukey's post-hoc test on logit-transformed data. (**B**) Number of lawn leaving events per animal in the same assays as (**A**). Statistics by Kruskal-Wallis test with Dunn's multiple comparisons test. (**C–D**) Animals on inedible food roam and leave lawns more than animals on edible food. (**C**) Fraction of time

*Figure 3 continued on next page*

*Figure 3 continued*

roaming on inedible bacteria generated by adding aztreonam dissolved in DMSO to *E. coli* growing on plates.+DMSO/+aztreonam n=64,+DMSO (control) n=66, Statistics by Student's t-test performed on logit-transformed data. (**D**) Number of lawn leaving events in the same assays as (**C**). Statistics by Mann-Whitney U-test. (**E–J**) Feeding inhibition by optogenetic depolarization of pharyngeal muscles stimulates roaming and lawn leaving. (**E**) Heatmap showing roaming and dwelling for animals before, during, and after 10 min optogenetic feeding inhibition. Data for animals pre-treated with all-trans retinal (+ATR) is shown. (**F**) Fraction of animals roaming before, during, and after optogenetic feeding inhibition. Light ON period denoted by yellow shading (+ATR n=36). Control animals not pre-treated with all-trans retinal (-ATR n=32). Statistics by Student's t-test comparing +/-ATR data averaged and logit-transformed during intervals indicated by black lines above plots: Minutes 0–10, 12–20, 22–30. (**G**) A greater fraction of animals leave lawns during feeding inhibition. Statistics by Fisher's exact test. (**H**) Number of lawn leaving events in the same assays as (**G**). Statistics by Wilcoxon rank-sum test. (**I**) Cumulative distribution of time until the first head poke reversal or lawn leaving event during feeding inhibition. Statistics by Kolmogorov-Smirnov two-sample test. (**J**) Roaming animals accelerate before leaving during feeding inhibition. Left, mean roaming speed of animals before leaving. Right, quantification of roaming speed increase from minutes –3 to –1 to the last 30 s before leaving. Statistics by Wilcoxon rank-sum test. Statistics: ns, not significant, * $p<0.05$, *** $p<10^{-3}$,**** $p<10^{-4}$ In time-averages (F,J), dark line represents the mean and shaded region represents the standard error. Violin plots show median and interquartile range. In (H), each dot pair connected by a line represents data from a single animal. In (J), each dot pair connected by a line represents data preceding a single lawn leaving event. Thick black line indicates the average. See *Figure 3—source data 1*.

The online version of this article includes the following source data and figure supplement(s) for figure 3:

**Source data 1.** Quantification of roaming and lawn leaving from experiments in *Figure 3*.

**Figure supplement 1.** Further quantification of roaming and leaving behaviors under food intake inhibition.

Optogenetic inhibition of feeding elicited immediate roaming and probabilistic lawn leaving in both *pdfr-1* and *tph-1* mutants (*Figure 4—figure supplement 1*), indicating that neither of these neuromodulators is essential for interpreting feeding inhibition. This was surprising because the *tph-1*-expressing NSM neurons sense bacterial ingestion and signal food availability via serotonin release (*Rhoades et al., 2019*), and were therefore candidates to relay feeding signals. NSM may act through additional transmitters as well as serotonin, or it may be redundant with additional neurons that detect feeding inhibition.

## Acute circuit manipulation drives deterministic roaming and probabilistic leaving

As a complement to the neuromodulatory mutants, we employed a circuit-based approach to manipulate arousal levels and examine effects on lawn leaving. Roaming is strongly stimulated by *pdfr-1*; we defined sites of *pdfr-1* expression that stimulate roaming using an intersectional Cre-Lox system that restores *pdfr-1* expression in targeted groups of cells in *pdfr-1* mutant animals (*Figure 5—figure supplement 1*, *Figure 5—figure supplement 2A*; *Flavell et al., 2013*). Previous work identified a moderate effect of AIY, RIM, and RIA neurons as mediators of *pdfr-1*-dependent roaming (*Flavell et al., 2013*). We observed a stronger rescue of roaming upon *pdfr-1* expression solely in the RIB neurons, which are active during rapid forward locomotion (*Figure 5—figure supplement 2B–D*; *Ji et al., 2021*; *Wang et al., 2020*).

Following this result, we asked whether optogenetic activation of RIB might be sufficient for roaming in wild-type animals. PDFR-1 signals through the heterotrimeric G protein Gαs to increase cAMP levels (*Janssen et al., 2008*), and optogenetic activation of groups of PDFR-1-expressing neurons with the bacterial light-activated adenylyl cyclase BlaC increases roaming (*Flavell et al., 2013*; *Ryu et al., 2010*). We found that acute optogenetic activation of only RIB with BlaC induced immediate roaming in over 80% of animals (*Figure 5A–C*, *Figure 5—figure supplement 3A–B*).

RIB::BlaC stimulation also potentiated lawn leaving, with over 70% of animals leaving lawns during the stimulation period (*Figure 5D–E*). Like lawn-leaving during optogenetic feeding inhibition, this behavior was probabilistic across the 10-min light pulse (*Figure 5F*). Animals accelerated slightly in the 30 s before leaving, but the speed increase appeared less pronounced than in other manipulations, suggesting that RIB activity may partially occlude the acceleration motif in lawn leaving (*Figure 5—figure supplement 3F*). Neither roaming nor lawn leaving was potentiated by blue light exposure in wild type control animals not expressing BlaC (*Figure 5C*, *Figure 5—figure supplement 3A, C-D*).

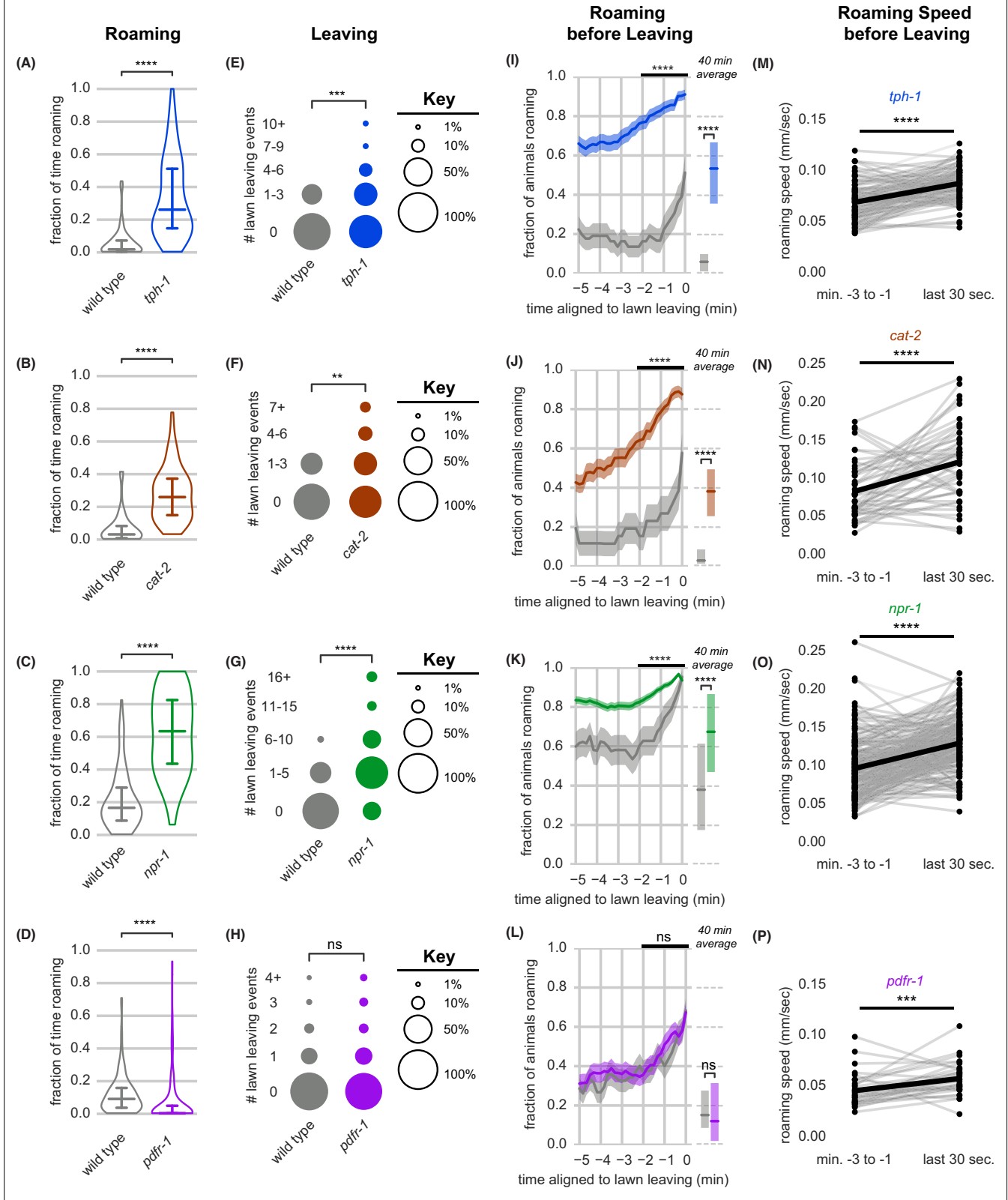

**Figure 4.** Neuromodulatory signaling mutants retain coupling of arousal and leaving. (**A–P**) Roaming, lawn leaving, roaming before leaving, and roaming speed quantified across mutants in four neuromodulatory genes known to alter roaming and dwelling: *tph-1*, *cat-2*, *npr-1*, and *pdfr-1*. (**A–D**) Fraction of time roaming. Statistics by Student's t-test on logit-transformed data. (**E–H**) Number of lawn leaving events per animal. Statistics by Mann-Whitney U test. (**I–L**) Fraction of animals roaming before lawn leaving. Left, fraction of animals roaming in the last 5 min before lawn leaving. WT and

*Figure 4 continued on next page*

*Figure 4 continued*

mutant roaming fractions were compared over the two minutes prior to leaving (black bar). Right, total fraction of time spent roaming and dwelling in all assays that included a lawn-leaving event. Statistics by Student's t-test on logit-transformed data. Note that roaming levels were unusually low (**A,B,I,J**) or high (**C,K**) in the wild-type controls for these groups, and therefore genotypes cannot be compared across different experimental panels. (**M–P**) Roaming speed before leaving computed at times when less than 10% of aligned traces had missing data. Paired plots indicate the average speed from minutes –3 to –1 and in the last 30 s before leaving per animal. Statistics by Wilcoxon rank-sum test. Statistics: ns not significant ($p>0.05$), ** $p<0.01$, *** $p<10^{-3}$, **** $p<10^{-4}$ Violin plots and box plots show median and interquartile range. In time-averages, dark line represents the mean and shaded region represents the standard error. In paired plots (M-P), each dot pair connected by a line represents data preceding a single lawn leaving event. Thick black line indicates the average. (*tph-1* n=139, wild type controls n=127; *cat-2* n=88, wild type controls n=76; *npr-1* n=91, wild type controls n=90; *pdfr-1* n=265, wild type controls n=247). See *Figure 4—source data 1*.

The online version of this article includes the following source data and figure supplement(s) for figure 4:

**Source data 1.** Quantification of roaming and lawn leaving from experiments in *Figure 4*.

**Figure supplement 1.** Optogenetic feeding inhibition stimulates roaming and leaving in neuromodulatory mutants.

## Chemosensory neurons couple roaming dynamics, internal state, and lawn leaving

In addition to the neuromodulatory arousal systems, multiple food- and pheromone-sensing chemosensory neurons affect roaming, dwelling, and leaving behaviors (*Supplementary file 1*). Most of these neurons use the cyclic nucleotide-gated channel gene *tax-4* for sensory transduction, and *tax-4* mutants have diminished roaming behavior (*Ben Arous et al., 2009*; *Fujiwara et al., 2002*; *Figure 6A*, *Supplementary file 1*). However, we found that *tax-4* mutant animals continued to leave lawns – indeed, they left at slightly higher rates than wild-type animals (*Figure 6B*, *Supplementary file 2*). Moreover, the temporal relationship between roaming and leaving was altered in *tax-4* mutants, which typically roamed for only ~1 min prior to lawn leaving (*Figure 6C*, *Figure 6—figure supplement 1A–B*). These results indicate that loss of *tax-4* disrupted the characteristic arousal dynamics associated with leaving.

Many of the 15 classes of sensory neurons that express *tax-4* have been implicated in roaming or leaving behaviors (*Supplementary file 1*). We rescued *tax-4* separately in AWC, which senses food odors; ASK, which senses amino acids and pheromones; ASJ and ASI, which sense pheromones, food, and toxins; and URX/AQR/PQR, which sense environmental oxygen (*Ben Arous et al., 2009*; *Bendesky et al., 2011*; *Greene et al., 2016*; *Milward et al., 2011*). Significant effects on roaming or leaving behavior were observed upon ASJ, ASK, or AWC rescue (*Figure 6A–C*, *Figure 6—figure supplement 1* and S2). The strongest effects resulted from *tax-4* rescue in the ASJ neurons, which partially restored roaming levels before lawn leaving (*Figure 6A–C*, *Figure 6—figure supplements 1 and 2*), and *tax-4* rescue in the ASK neurons, which paradoxically suppressed leaving to a level below that of either wild type or *tax-4* animals (*Figure 6—figure supplement 2*). Because ASJ and ASK showed opposite effects in these experiments, we also examined strains in which both neurons were rescued. Combined ASJ and ASK rescue normalized roaming and leaving compared to ASK rescue alone, albeit not to fully wild-type levels (*Figure 6—figure supplement 2*). While we have not tested all neurons and combinations, these results suggest that multiple *tax-4*-expressing sensory neurons have roles in arousal-related behaviors and highlight ASJ as a regulator of roaming and leaving dynamics.

Next, we asked how *tax-4* mutants responded to acute inhibition of feeding. As in wild-type animals, optogenetic inhibition of pharyngeal pumping resulted in an immediate and strong increase in roaming in *tax-4* mutants (*Figure 6D*, *Figure 6—figure supplement 3A*). However, feeding inhibition in *tax-4* mutants increased leaving only slightly, unlike in the wild-type (*Figure 6E*, *Supplementary file 2*). Both of these effects were rescued by expressing *tax-4* in ASJ neurons (*Figure 6C–E*, *Figure 6—figure supplement 3B–D*). Similarly, *tax-4* animals on inedible food roamed at the same high rate as wild type animals but produced significantly fewer lawn leaving events (*Figure 6—figure supplement 4*); rescuing *tax-4* in ASJ restored lawn leaving to wild-type levels. Thus *tax-4* sensory mutants uncouple leaving behavior from its normal context in multiple ways: they can leave edible food lawns without an extended roaming state, and they are less likely to leave when feeding is inhibited, even while they are roaming.

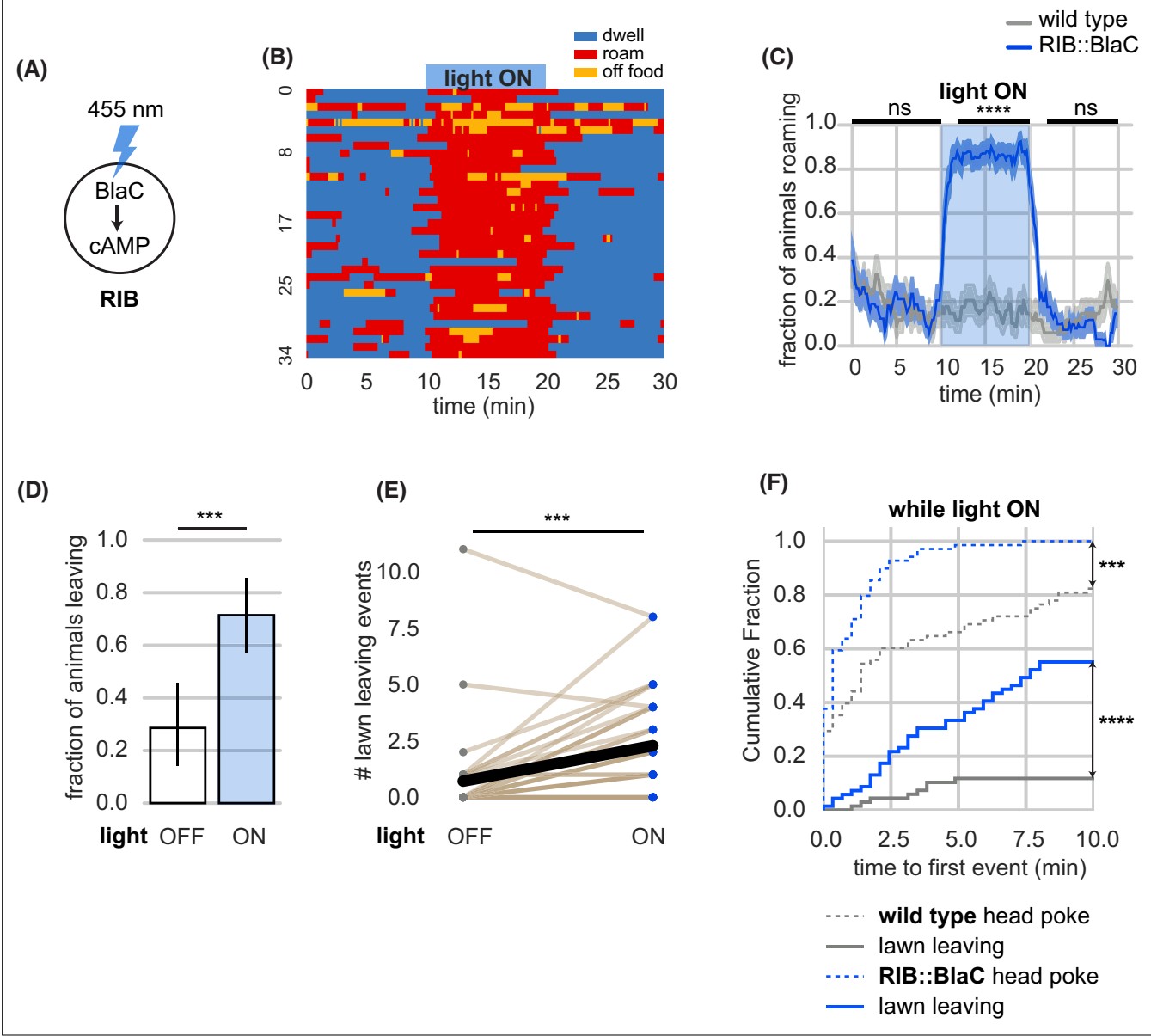

**Figure 5.** Acute circuit manipulation drives deterministic roaming and probabilistic leaving. (**A**) Experimental design. Stimulation of bacterial light-activated adenylyl cyclase (BlaC) with 455 nm light increases cAMP synthesis in RIB::BlaC neurons. (**B**) Heatmap showing roaming and dwelling for RIB::BlaC animals before, during, and after 10-min optogenetic stimulation. (**C**) Fraction of animals roaming before, during, and after optogenetic blue light stimulation. Light ON period denoted by blue shading. Control animals are wild type. Statistics by Student's t-test comparing RIB::BlaC and wild-type animals averaged and logit-transformed during intervals indicated by black lines above plots: Data compared at 0–10, 12–20, 22–30 min. (**D**) A greater fraction of RIB::BlaC animals leave lawns when the light is ON vs. OFF. Statistics by Fisher's exact test. (**E**) Number of lawn leaving events under the same conditions as (**D**). Statistics by Wilcoxon rank-sum test. (**F**) Cumulative distribution of time until the first head poke reversal or lawn leaving event while the light is ON. Statistics by Kolmogorov-Smirnov two-sample test. Statistics: ns not significant (p>0.05), ** p<0.01, *** p<10⁻³, **** p<10⁻⁴ In time-averages (C), dark line represents the mean and shaded region represents the standard error. (RIB::BlaC n=35, wild type n=34) In (E), each dot pair connected by a line represents data from a single animal. Thick black line indicates the average. See *Figure 5—source data 1*.

The online version of this article includes the following source data and figure supplement(s) for figure 5:

**Source data 1.** Quantification of roaming and lawn leaving from experiments in *Figure 5*.

**Figure supplement 1.** *pdfr-1* genomic characteristics and expression patterns.

**Figure supplement 2.** Transgenic rescue of *pdfr-1* in RIB neurons restores roaming (**A**) Schematic depicting intersectional cell-specific rescue of *pdfr-1* using an inverted Cre-Lox strategy.

**Figure supplement 3.** Further quantification and controls of RIB::BlaC experiments.

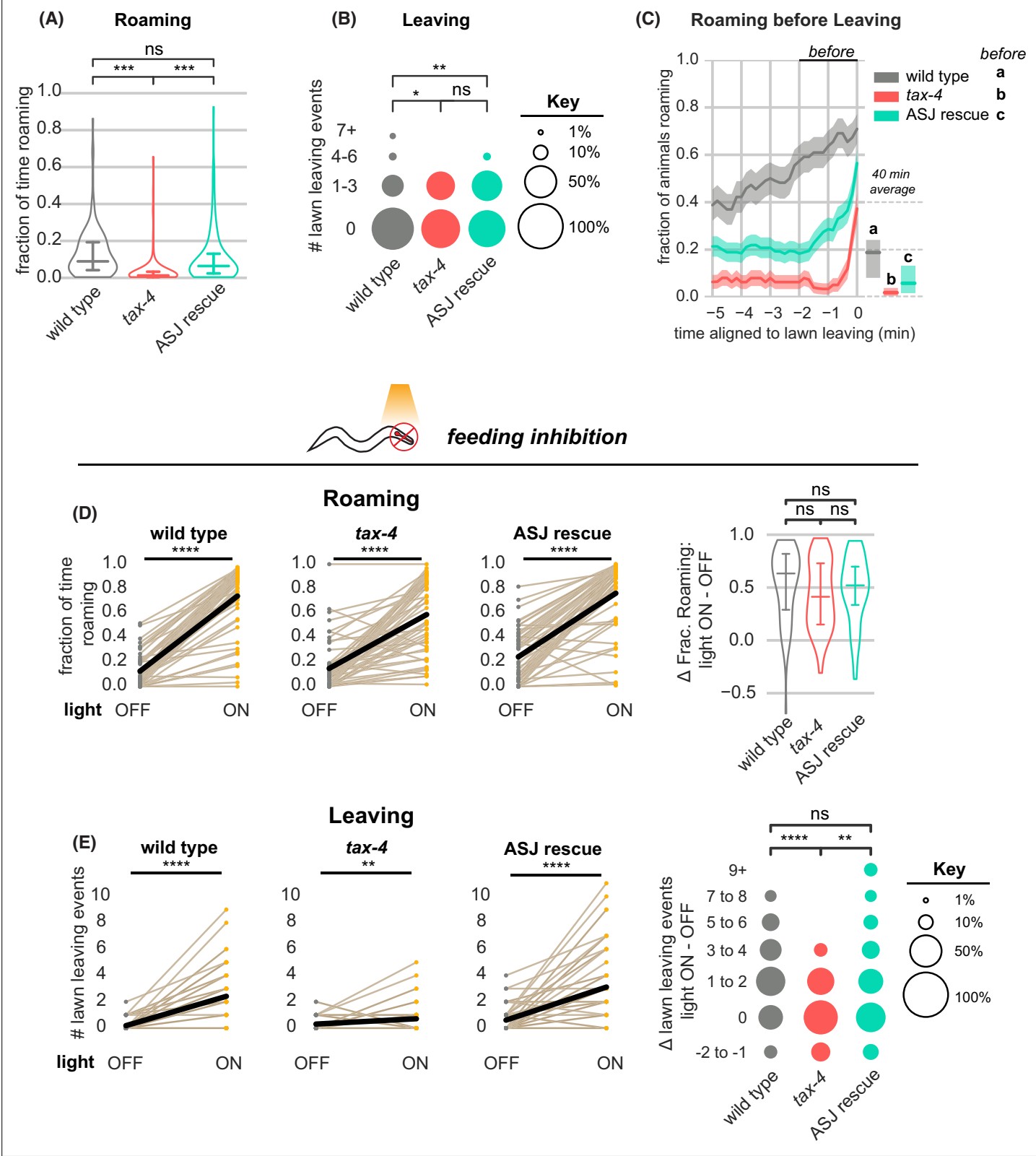

**Figure 6.** *tax-4*-expressing sensory neurons couple roaming and lawn leaving. (**A–C**) Roaming and leaving in *tax-4* mutants, and rescue by *tax-4* expression in ASJ neurons (wild type n=143, *tax-4* n=156, *tax-4* ASJ rescue n=148). Additional features of roaming and leaving are shown in *Figure 6— figure supplement 1*; results with additional rescued neurons are in *Figure 6—figure supplement 2*. (**A**) Fraction of time roaming. Statistical tests, one-way ANOVA followed by Tukey's post hoc test on logit-transformed data. (**B**) Number of lawn-leaving events per animal. Statistical tests, Kruskal-

*Figure 6 continued on next page*

*Figure 6 continued*

Wallis followed by Dunn's multiple comparisons test. (**C**) Fraction of animals roaming before lawn-leaving. Left, fraction of animals roaming in the last 5 min before lawn leaving. Roaming fractions were compared over the 2 min prior to leaving (black bar). Right, total fraction of time spent roaming and dwelling in all assays that included a lawn-leaving event. Key, statistical tests of differences in roaming two minutes before leaving by one-way ANOVA followed by Tukey's post hoc test on logit-transformed data. (**D–E**) Optogenetic feeding inhibition in *tax-4* mutants and rescued strains (wild type n=70, *tax-4* n=66, *tax-4* ASJ rescue n=68). (**D**) Left, Fraction of time spent roaming under optogenetic feeding inhibition in *tax-4* and ASJ rescue strain. Statistics by paired t-test on logit-transformed data. Right, Difference in the fraction of time roaming when the light is ON and OFF. Statistics by one-way ANOVA followed by Tukey's post hoc test on logit-transformed data. (**E**) Left, Number of lawn leaving events under optogenetic feeding inhibition in *tax-4* and ASJ rescue strain. Statistics by Wilcoxon rank-sum test. Right, Difference in the number of lawn leaving events when the light is ON and OFF. Statistics by Kruskal-Wallis test followed by Dunn's multiple comparisons test. ns not significant (p>0.05), * p<0.05, *** $p<10^{-3}$, **** $p<10^{-4}$ Violin plots and box plots (A,D) show median and interquartile range. In time-averages (C), dark line represents the mean and shaded region represents the standard error. In paired plots (D,E), each dot pair connected by a line represents data from a single animal. Thick black line indicates the average. See *Figure 6—source data 1*.

The online version of this article includes the following source data and figure supplement(s) for figure 6:

**Source data 1.** Quantification of roaming and lawn leaving from experiments in *Figure 6*.

**Figure supplement 1.** Further quantification of *tax-4* mutants and *tax-4* ASJ rescue on edible food (**A**) Complementary cumulative distribution functions (ccdfs) for overall roam state durations for wild type, *tax-4,* and *tax-4* rescued in ASJ.

**Figure supplement 2.** *tax-4* rescue in additional *tax-4*-expressing neurons (**A–B**) Quantification of roaming and leaving in *tax-4* mutants with *tax-4* rescued in ASJ, ASK or ASJ +ASK neurons (wild type n=95, *tax-4* n=104, ASJ rescue n=91, ASK rescue n=101, ASJ +ASK rescue n=105).

**Figure supplement 3.** Further quantification of *tax-4* mutants and *tax-4* rescue upon optogenetic feeding inhibition (**A-D**) Effects of *tax-4* on roaming and leaving behavior induced by optogenetic feeding inhibition.

**Figure supplement 4.** *tax-4* mutants leave less on inedible food and ASJ rescue restores leaving.

## Discussion

Both in the wild and in laboratory settings, foraging locomotion patterns exhibit scale invariance, meaning that similar statistics of movement displacement and duration arise on short and long timescales (*Ayala-Orozco et al., 2004*; *Proekt et al., 2012*). Long-term arousal changes are regulated by neuromodulators that signal widely throughout the brain (*Flavell et al., 2022*; *Taghert and Nitabach, 2012*; *Weissbourd et al., 2014*), whereas moment-by-moment decision-making involves fast sensorimotor circuits (*Gold and Shadlen, 2007*). Here, we show that *C. elegans* couples food leaving decisions, which unfold over seconds, to high arousal roaming states that last minutes. Although food intake and neuromodulatory signaling both alter the frequency of sustained *C. elegans* roaming states, these changes do not disrupt the coupling of roaming to food leaving decisions. Instead, sensory neurons link roaming and leaving behaviors, integrating these behavioral motifs, their dynamic properties, and their regulation by food intake.

### Behavioral arousal alters probabilistic decision-making

From an ethological perspective, lawn-leaving is a classical foraging decision based on an animal's assessment of the quality of a food patch (*Charnov, 1976*). Previous studies of lawn-leaving have identified features of the environment, like food availability, population density, and toxic repellents, that affect leaving probability as predicted by foraging theory. Here, we have extended these observations using the framework of computational neuroethology (*Datta et al., 2019*), examining behavior in detail across time to identify rare leaving events and the behavioral context in which they occurred. We found that leaving was tightly coupled to roaming both during spontaneous behaviors and after acute manipulations that induced roaming states. In each case, and in neuromodulatory mutants that altered roaming frequency, leaving occurred when animals were roaming, occurred probabilistically over minutes during the roaming state, and was preceded by a brief 30 s acceleration in speed. The stereotyped features of leaving behavior, which were not shared by other lawn edge encounters, suggest that it is a discrete behavioral motif associated with roaming states. Leaving can be viewed as a form of decision-making—a discrete action that drives an adaptive behavioral choice between alternatives, influenced by internal states such as arousal (*Kennedy et al., 2014*).

Roaming and dwelling comprise a widely used framework for defining arousal states in *C. elegans*, although changing experimental conditions or analysis methods can reveal sub-states or alternative classification frameworks (this work and *Cermak et al., 2020*; *Gallagher et al., 2013*; *Raizen et al.,*

*2008*; *Van Buskirk and Sternberg, 2007*). Classical ethology presents such internal states as hierarchical and mutually exclusive (*Tinbergen, 1951*), while molecular and circuit analysis has extended and refined these ideas (*Asahina et al., 2014*; *Flavell et al., 2013*; *Hindmarsh Sten et al., 2021*; *Hong et al., 2014*; *Moy et al., 2015*; *Pan et al., 2012*). Our results indicate that lawn leaving behaviors are coupled to the roaming state, but within this state they are relatively rare and apparently probabilistic. More detailed studies may reveal additional regulators, such as an individual's sensory experience, that shape the pattern of leaving behavior (*Coen et al., 2014*).

## Sensory neurons integrate internal state and external sensation to guide foraging decisions

Our results identify two distinct biological mechanisms that regulate lawn leaving. First, leaving is coupled to arousal state. Leaving rates are 20-fold higher in roaming animals than in dwelling animals, and the leaving rates of most arousal mutants are largely explained by the fact that they spend more time roaming (*Supplementary file 2*). The second mechanism is controlled by *tax-4* sensory neurons, which shape the behavioral dynamics of roaming and leaving across a range of conditions and stimulate leaving during feeding inhibition.

At a straightforward level, sensory neurons are well-placed to evaluate environmental quality in the framework of foraging theory. For example, ASK and AWC neurons sense amino acids and food odors (*Bargmann, 2006*), while ASJ neurons sense pheromones and toxins (*Greene et al., 2016*; *Meisel et al., 2014*), and other sensory neurons detect distinct environmental features (*Bargmann, 2006*). Accordingly, many sensory neurons affect roaming, dwelling, or leaving (*Supplementary file 1*), and their relative importance varies with context, such as the presence of chemical repellents or pheromones that signal population density (*Dal Bello et al., 2021*; *Greene et al., 2016*; *Milward et al., 2011*; *Pradel et al., 2007*). Under the conditions used here, ASK neurons suppressed and ASJ neurons promoted roaming and leaving, respectively. Rescuing *tax-4* in ASJ partially restored features of normal leaving dynamics, including roaming before leaving.

Others have shown that chronic feeding deprivation across hours drives lawn leaving through the action of *tax-4* in several sensory neurons (*Olofsson, 2014*). An unexpected result obtained here was that *tax-4* sensory neurons were required to drive the high leaving rates after acute inhibition of feeding. Together, these results suggest that the sensory neurons are sites at which internal feeding cues, behavioral states, and specific foraging decisions are integrated. Additional experiments are needed to define the full set of sensory neurons that couple feeding inhibition to leaving, the nature of the interoceptive signal from pharyngeal pumping, and the readout of the sensory neurons. Pharyngeal interoception is mediated in part by the serotonergic NSM neurons (*Rhoades et al., 2019*), but we found that serotonin was not essential for the effects of acute feeding inhibition. With respect to interoception, some of the *tax-4*-expressing sensory neurons detect tyramine, one of several biogenic amines that regulate lawn leaving (*Bendesky et al., 2011*; *O'Donnell et al., 2020*), and many detect neuropeptides as well (*Hapiak et al., 2013*).

Sensory neurons in *C. elegans* and other animals are most often studied in the context of rapid behavioral responses, but they also have critical roles in endocrine signaling. For example, intrinsically photosensitive retinal ganglion cells drive circadian entrainment in mammals (*Hattar et al., 2002*) and in cichlid fish, visual stimuli from females are sufficient to induce an endocrine androgen response in males (*O'Connell et al., 2013*). We speculate that the combination of endocrine signals, neuropeptides, and fast transmitters released by sensory neurons couples roaming states to leaving states. The insulin peptides and DAF-7-TGF-beta protein that regulate roaming, dwelling, and leaving are primarily produced by *tax-4*-expressing sensory neurons, so these neurons bridge sensory and endocrine regulation of foraging (*Taylor et al., 2021*). Expression of these genes is regulated by pheromones and metabolic conditions, providing an additional layer beyond neural activity for long-term regulation of behavioral state. Many *tax-4* sensory neurons also release classical neurotransmitters such as glutamate (ASK, AWC) and acetylcholine (ASJ), and therefore have the potential to modulate rapid behaviors such as the lawn-leaving motif (*Taylor et al., 2021*). Future studies can test this hypothesis while determining the mechanisms by which sensory neurons integrate signals to trigger lawn leaving.

# Methods

**Key resources table**

| Reagent type (species) or resource | Designation | Source or reference | Identifiers | Additional information |
|---|---|---|---|---|
| Strain, strain background (*Caenorhabditis elegans*, N2 hermaphrodite) | Wild type | **Yoshimura et al., 2019** | PD1074 (now CGC1) | Used in all figures except **Figure 4** *tph-1* and *cat-2* controls |
| Strain, strain background (*Caenorhabditis elegans*, N2 hermaphrodite) | Wild type | This paper | ID_BargmannDatabase:CX0001 | Used as wild type controls in **Figure 4** *tph-1* and *cat-2* experiments |
| Strain, strain background (*Caenorhabditis elegans*, N2 hermaphrodite) | pharynx ReaChR | This paper | ID_BargmannDatabase:CX16279 | **Figures 3 and 6** and supplements |
| Strain, strain background (*Caenorhabditis elegans*, N2 hermaphrodite) | *pdfr-1* | **Flavell et al., 2013** | ID_BargmannDatabase:CX14295 | **Figure 4** |
| Strain, strain background (*Caenorhabditis elegans*, N2 hermaphrodite) | *tph-1* | https://cgc.umn.edu/strain/MT15434 | ID_BargmannDatabase:MT15434 | **Figure 4** |
| Strain, strain background (*Caenorhabditis elegans*, N2 hermaphrodite) | *cat-2* | **Stern et al., 2017** | ID_BargmannDatabase:CX11078 | **Figure 4** |
| Strain, strain background (*Caenorhabditis elegans*, N2 hermaphrodite) | *npr-1* | **Jang et al., 2017** | ID_BargmannDatabase:CX13663 | **Figure 4** |
| Strain, strain background (*Caenorhabditis elegans*, N2 hermaphrodite) | pharynx ReaChR; *pdfr-1* | This paper | ID_BargmannDatabase:CX16528 | **Figure 4—figure supplement 1** |
| Strain, strain background (*Caenorhabditis elegans*, N2 hermaphrodite) | pharynx ReaChR; *tph-1* | This paper | ID_BargmannDatabase:CX16529 | **Figure 4—figure supplement 1** |
| Strain, strain background (*Caenorhabditis elegans*, N2 hermaphrodite) | RIB::BlaC | This paper | ID_BargmannDatabase:CX18471 | **Figure 5, Figure 5—figure supplement 3** |
| strain, strain background (*Caenorhabditis elegans*, N2 hermaphrodite) | PCR fragment #1; *pdfr-1* | **Flavell et al., 2013** | ID_BargmannDatabase:CX14378 | **Figure 5—figure supplement 1** |
| Strain, strain background (*Caenorhabditis elegans*, N2 hermaphrodite) | PCR fragment #2; *pdfr-1* | **Flavell et al., 2013** | ID_BargmannDatabase:CX14383 | **Figure 5—figure supplement 1** |
| Strain, strain background (*Caenorhabditis elegans*, N2 hermaphrodite) | *pdfr-1:: CreONpdfr-1* | **Flavell et al., 2013** | ID_BargmannDatabase:CX14485 | **Figure 5—figure supplement 2** |
| Strain, strain background (*Caenorhabditis elegans*, N2 hermaphrodite) | RIB *pdfr-1* rescue; *pdfr-1* | This paper | ID_BargmannDatabase:CX18302 | **Figure 5—figure supplement 2** |
| Strain, strain background (*Caenorhabditis elegans*, N2 hermaphrodite) | pan-neuronal *pdfr-1* rescue; *pdfr-1* | **Flavell et al., 2013** | ID_BargmannDatabase:CX14488 | **Figure 5—figure supplement 2** |
| Strain, strain background (*Caenorhabditis elegans*, N2 hermaphrodite) | AIY, RIM, RIA *pdfr-1* rescue; *pdfr-1* | **Flavell et al., 2013** | ID_BargmannDatabase:CX14271 | **Figure 5—figure supplement 2** |
| Strain, strain background (*Caenorhabditis elegans*, N2 hermaphrodite) | *tax-4* | **Worthy et al., 2018** | ID_BargmannDatabase:CX13078 | **Figure 6, Figure 6—figure supplements 1 and 2**, 4 |

*Continued on next page*

*Continued*

| Reagent type (species) or resource | Designation | Source or reference | Identifiers | Additional information |
|---|---|---|---|---|
| Strain, strain background (*Caenorhabditis elegans*, N2 hermaphrodite) | ASJ rescue; *tax-4* | This paper | ID_BargmannDatabase:CX11118 | *Figure 6, Figure 6—figure supplements 2 and 4* |
| Strain, strain background (*Caenorhabditis elegans*, N2 hermaphrodite) | ASK rescue; *tax-4* | This paper | ID_BargmannDatabase:CX13361 | *Figure 6—figure supplement 2* |
| Strain, strain background (*Caenorhabditis elegans*, N2 hermaphrodite) | ASJ +ASK rescue; *tax-4* | This paper | ID_BargmannDatabase:CX11110 | *Figure 6—figure supplement 2* |
| Strain, strain background (*Caenorhabditis elegans*, N2 hermaphrodite) | AWC rescue; *tax-4* | *Worthy et al., 2018* | ID_BargmannDatabase:CX13790 | *Figure 6—figure supplement 2* |
| Strain, strain background (*Caenorhabditis elegans*, N2 hermaphrodite) | ASI rescue; *tax-4* | This paper | ID_BargmannDatabase:CX11558 | *Figure 6—figure supplement 2* |
| Strain, strain background (*Caenorhabditis elegans*, N2 hermaphrodite) | URX/AQR/PQR rescue; *tax-4* | This paper | ID_BargmannDatabase:CX11113 | *Figure 6—figure supplement 2* |
| Strain, strain background (*Caenorhabditis elegans*, N2 hermaphrodite) | pharynx ReaChR; *tax-4* | This paper | ID_BargmannDatabase:CX18452 | *Figure 6, Figure 6—figure supplement 3* |
| Strain, strain background (*Caenorhabditis elegans*, N2 hermaphrodite) | pharynx ReaChR; *tax-4*; ASJ rescue | This paper | ID_BargmannDatabase:CX18538 | *Figure 6, Figure 6—figure supplement 3* |
| Strain, strain background (*Escherichia coli*, OP50) | OP50 | *Caenorhabditis* Genetics Center (CGC) | https://cgc.umn.edu/strain/OP50 | |
| Chemical compound, drug | Aztreonam | Sigma | PZ0038 | CAS: 78110-38-0 |
| Chemical compound, drug | all trans-Retinal (ATR) | Sigma | R2500 | CAS: 116-31-4 |
| Software, algorithm | ImageJ | ImageJ (https://imagej.nih.gov/) | RRID:SCR_003070 | Version 1.50i |
| Software, algorithm | MATLAB | Mathworks (https://www.mathworks.com/) | RRID:SCR_001622 | Version R2018a, R2020a, R2021a, R2023a |
| Software, algorithm | FlyCapture | Pointgrey (https://www.ptgrey.com/) | | Version FlyCap2 |
| Software, algorithm | Python | Python (python.org) | RRID:SCR_008394 | Version 3.8.3 |
| Software, algorithm | tracking and analysis code | this paper; *Scheer and Bargmann, 2023* | https://github.com/BargmannLab/Scheer_Bargmann2023 | |

## Nematode and bacterial culture

Bacterial food used in all experiments was *E. coli* strain OP50. Nematodes were grown at 20 °C on nematode growth media plates (NGM; 51.3 mM NaCl, 1.7% agar, 0.25% peptone, 1 mM CaCl2, 12.9 µM cholesterol, 1 mM MgSO₄, 25 mM KPO4, pH 6) seeded with 200 µL of a saturated *E. coli* liquid culture that had been grown at room temperature for 48 hr or overnight at 37 °C (without shaking) from a single colony of OP50 in 100 mL of sterile LB (*Brenner, 1974*). All experiments were performed on young adult hermaphrodites, picked as L4 larvae the evening before an experiment. Wild-type controls were the CGC1 (previously PD1074) sequenced strain derived from the N2 Bristol strain (*Yoshimura et al., 2019*), except for *Figure 4 tph-1* and *cat-2* controls, which were the CX0001 isolate of the N2 Bristol strain. Mutant strains were backcrossed into wild type to reduce background mutations. Transgenic strains were always compared to matched controls tested in parallel on the

same days. Full genotypes and detailed descriptions of all strains and transgenes appear in *Supplementary file 3*: Strain details.

## Molecular biology and transgenics

Strains tested for *pdfr-1* rescue using PCR-amplified genomic fragments (*Figure 5—figure supplement 1*) were from *Flavell et al., 2013*. For cell-selective *pdfr-1* rescue, an inverted cDNA under the *pdfr-1* distal promoter was the floxed rescue construct (*Figure 5—figure supplement 2*), and Cre expression was driven by a pan-neuronal promoter (*tag-168*), in RIM (*tdc-1*), in AIY and others (*mod-1*), in RIA (*glr-3),* and in RIB (*sto-3*). A 972 bp region upstream of the *sto-3* gene that drives expression solely in the RIB neurons was cloned using these primers:

> *sto-3:* gatgcccaatcagtttttttttcaccaa, aagccaaaccaagtgagaagaagtattca

Strains and extrachromosomal arrays for *tax-4* rescue were reported in *Macosko et al., 2009* and *Worthy et al., 2018*. The sequences of the promoter ends are given below, along with the concentration at which the plasmid was injected for generating transgenic lines.

> ASJ *tax-4* rescue *srh-11*: gggcaaggacaatgttgccgcag, tgggaataaaataacgacgtatgaata, 50 ng/µl
> ASK *tax-4* rescue
> *sra-9*: gcatgctatattccaccaaaagaaa, tagcttgtgcatcaatcatagaaca 50 ng/µl
> AWC *tax-4* rescue *ceh-36*: ctcacatccatctttctggcgact, ttgtgcatgcgggggcaggcga, 30 ng/µl
> ASI *tax-4* rescue *str-3*: gtgaacttgaaaagcgcaagtgatat, ttccttttgaaattgaggcagttgtc, 100 ng/µl
> URX/AQR/PQR *tax-4* rescue
> *gcy-36*: tggatgttggtagatggggtttgga, aaattcaaacaagggctacccaaca 2 ng/µl

Transgenic animals were generated by microinjection of DNA containing the genetic construct of interest, a fluorescent co-injection marker (*myo-2p*::mCherry, *myo-3p*::mCherry, *elt-2p*::nGFP, *elt-2p*::mCherry), and empty pSM vector to reach a final DNA concentration of 100 ng/µL. Transgenes were maintained as extrachromosomal arrays.

## Small lawn foraging assay

For all assays, *E. coli* OP50 was grown overnight in a shaking LB liquid culture from a single colony at 37 °C. On the morning of the assay, 400 µL of saturated liquid culture was diluted into 5 mL of LB and allowed to grow to OD1 at 37 °C (~1.5 hours), as measured by spectrophotometer. The liquid culture was then spun down and resuspended in M9 buffer (3 g $KH_2PO_4$, 6 g $Na_2HPO_4$, 5 g NaCl, 1 ml 1 M $MgSO_4$, $H_2O$ to 1 liter) then concentrated to a density of OD2 (and OD1 or OD4 in *Figure 3A–B*). To generate the test lawns, 2 µL of this concentrated bacterial resuspension was seeded onto NGM agar in the center of each well of a custom-made laser-cut six-well plate, where each well is 10 mm in diameter (*Stern et al., 2017*).

50 µL of bacterial resuspension was seeded onto a separate NGM agar plate to be used as a food density acclimation plate. Lawns were grown at 20–22°C for 2 hr before the assay. Adult hermaphrodites picked as L4s 16–20 hr before the assay were then transferred to acclimation plates. After 45–90 min, animals were transferred to an unseeded NGM plate, cleaned of *E. coli*, and transferred singly into each well of the assay plates, where they were placed on bacteria-free agar and allowed to find the small food lawn on their own. Animals of the same genotype were grouped on the same six-well plates and each plate was recorded by a single camera. We used 12 cameras, enabling simultaneous recording of up to 72 individual animals at a time. Temperature and relative humidity within the behavioral recording apparatus were continuously monitored during recordings to ensure that environmental conditions were consistent across filming locations. As a further precaution, the filming locations of each genotype and wild type controls within the recording apparatus were randomized across batches of experiments and days to average out behavioral influences deriving from non-uniform local environmental conditions. Assays were recorded for 1 hr at 3 frames per second using 12 8.8 MP USB3 cameras (Pointgrey, Flea3) and 35 mm high-resolution objectives (Edmund Optics). LED backlights (Metaphase Technologies) provided uniform illumination of the assay plates. Commercial software (Flycapture, Pointgrey) was used to record the movies.

## Uniform lawn assay

Assays testing worm behavior on uniform bacterial lawns were performed as in the small lawn foraging assay with the exception that instead of 2 μL of bacteria seeded in the center of the well, 15 μL of OD2 bacterial suspension was spread evenly throughout the well. Bacteria were grown for 2 hr before acclimation and starting the assay.

## Optogenetic feeding inhibition assay

Pharyngeal pumping was inhibited by expressing the red-shifted channelrhodopsin ReAChR (*Lin et al., 2013*) under the *myo-2* pharyngeal muscle promoter (a strain generously provided by Steve Flavell). Animals were stimulated while navigating small food lawns described above. Experimental animals were grown on bacterial lawns containing 50 μM all-trans retinal (+ATR) overnight before assays. Control animals were placed on lawns made in parallel that did not contain retinal (-ATR). A 590 nm Precision LED with Uniform Illumination (Mightex) controlled with custom MATLAB software was used to deliver optogenetic stimuli. Animals were acclimated to small lawns (no ATR) for 20 min before being exposed to alternating 10 min intervals of light OFF and light ON using 590 nm light at 60 μW/mm$^2$ strobed at 10 Hz with a 50% duty cycle (*Figure 3*, *Figure 4—figure supplement 1*, *Figure 6*, *Figure 6—figure supplement 3*). Recording hardware and software was identical to that of off-food foraging assays without optogenetic stimulation except that 475 nm short-pass filters were used on recording optics (Edmund Optics) to prevent overexposure of the video recording during light pulse delivery.

For behavioral quantification in *Figure 3E–F*, *Figure 4—figure supplement 1A and C–D*, and *Figure 6—figure supplement 3A*, only data before, during and after the first light pulse is shown. For aggregate comparisons of within-animal behavior in light OFF vs. light ON conditions, the two light OFF and light ON pulses were merged, that is merging intervals 0–10 min +20–30 min, and 10–20 min + 30–40 min for statistical analyses. For statistical tests comparing the fraction of animals roaming across genotypes or conditions, steady state light intervals were used for averaging and comparing as indicated in the figure legends: 0–10, 12–20, 22–30 min.

## RIB::BlaC assay

Activation of the RIB neurons was accomplished using blue light-activated adenylyl cyclase BlaC (*Ryu et al., 2010*). BlaC was cloned into the pSM vector under the *sto-3* promoter to drive expression in the RIB neurons. Stimulation was carried out as described for ReAChR (above), except no all-trans retinal was applied and the animals were exposed to blue light (455 nm) at 3 μW/mm$^2$ strobed at 10 Hz with a 50% duty cycle (*Figure 5*, *Figure 5—figure supplement 3*). To prevent overexposure of the video recording during light pulse delivery, 525-nm long pass filters were used on recording optics (Edmund Optics). Behavioral quantification for *Figure 5B–C* was conducted as above (optogenetic feeding inhibition assay).

## Inedible food assay

Experiments with inedible bacterial food generated by addition of aztreonam were performed following the protocol of *Gruninger et al., 2008*. *E. coli* OP50 was grown in LB from a single colony to saturation overnight. On the morning of the assay, 400 μL of saturated liquid culture was diluted into 5 mL of LB with aztreonam (10 μg/ml) and allowed to grow to OD1 at 37 °C. To generate test lawns, 2 μL of this bacterial suspension was plated on NGM agar test plates containing aztreonam (10 μg/ml). Small lawn assays were performed as described above (see 'Small lawn foraging assay'), except that bacteria on NGM +aztreonam plates were allowed to grow for 4 hr before assay testing. To test for any acute behavioral responses to aztreonam, we also performed a control 'post-add' experiment, in which 2 μL of either 4 μg/mL aztreonam dissolved in DMSO or DMSO alone was added to normal small bacterial lawns after 4 hr of growth on NGM agar (*Figure 3—figure supplement 1*).

## Behavioral tracking and lawn feature detection

Because each video recorded the behavior of up to 6 individual animals, videos were manually cropped so each surrounded just a single animal using FFmpeg software (*Tomar, 2006*). To extract animal positions and postures, captured movies were analyzed by custom made scripts in MATLAB (Mathworks, version 2021a) using the Image Processing Toolbox and the Computer Vision Toolbox. In each frame

of the movie, the worm is segmented by background subtraction and its XY position is tracked over time using a Kalman filter. From the background-subtracted worm image, a smooth spline of 49 points was computationally applied and features relating to the movement of points along the body were derived following *Javer et al., 2018*. Disambiguation of the head versus tail was determined by assigning the head as the end of the spline that had greater cumulative displacement over the video assay, facilitating determination of times when the animal moved forward and backward.

## Behavioral features extracted

Features were defined as described in *Javer et al., 2018*. Briefly, body parts were defined based on a skeletonized spline containing 49 points equally distributed along the length of the worm body. The head comprises spline points 1–8. The midbody comprises spline points 17–33. The positions of head and midbody were calculated by deriving the centroid of each of these point sets by averaging their x and y positions before subsequent analyses (see *Figure 2—figure supplement 2*).

- *Midbody speed:* the derivative of displacement of the midbody across frames. Positive and negative numbers indicate forward and backward motion, respectively. Units: mm/sec.
- *Midbody angular speed:* The angle change across midbody positions over time: Two vectors are measured: $v_{0-1}$, representing the change in midbody position from time frame 0–1, and $v_{1-2}$, representing the change in midbody position from time frame 1–2. the position change over three frames at each time point was quantified. Angular speed is defined as the arc-cosine of the dot product of $v_{0-1}$ and $v_{1-2}$ divided by the scalar product of the norms of these vectors. Units: degrees per second.
- *Head speed:* the derivative of displacement of the head across frames. Units: mm/second.
- *Head angular speed:* Same as midbody angular speed but calculated for the head position. Units: degrees per second.
- *Head radial velocity relative to the midbody*: The derivative of displacement of the head relative to the midbody across frames. This is calculated by subtracting the midbody position from the head position and shifting to polar coordinates ($\Phi$,r) where $\Phi$ is the angular dimension and r is the radial dimension. Head radial velocity relative to midbody is the derivative of r with respect to time. Units: mm/s.
- *Head angular velocity relative to the midbody:* the derivative of the angular displacement of the head relative to the midbody across frames. The same procedure as above is used to generate polar coordinates. Head angular velocity relative to the midbody is the derivative of $\Phi$ with respect to time. Units: degrees per second.
- *Quirkiness:* $Q = \sqrt{1 - \frac{a^2}{A^2}}$, where a is the minor and A is the major axis of a bounding box surrounding the animal, respectively. Values closer to 1 means the animal's shape is more elongated and thinner; closer to 0 indicates a more rounded shape.

All these features were binned into contiguous 10 s intervals for subsequent analyses. There are also several features only defined for 10 s bins:

- *Fraction of time moving forward* per bin.
- *Fraction of time moving reverse* per bin.
- *Fraction of time paused* per bin.
- *Midbody forward speed*, the mean of Midbody speed where values are $\geq 0$.
- *Midbody reverse speed*, the mean of Midbody speed where values are $\leq 0$.

## Lawn boundary-related metrics and Lawn boundary interaction behaviors

The outline of the bacterial lawn was determined by edge detectors applied to the background averaged across movie frames. Across every frame, the closest boundary point to the animal's head was determined and used to calculate 'lawn boundary distance'.

Lawn boundary distance and movement direction were used to classify a set of lawn boundary interaction behaviors. Head pokes were classified based on an excursion of the head that peaks outside the lawn before returning to the lawn interior. In the period following maximal displacement outside the lawn and before resuming locomotion inside the lawn ('recovery interval'), three types of head pokes were categorized: head poke forward, in which at least half of the recovery interval is spent moving forward, head poke reversal, in which the animal executes a reversal during the recovery

interval, or head poke pauses, in which an animal spends at least half of the recovery interval with speed less than 0.02 mm/s. Lawn leaving events were marked as the first frame when the animal's head emerged from the lawn before its entire body exited the lawn.

## Quality control for including animals in subsequent analyses

Behavior and lawn features were detected and tracked over the 1 hr assay but only the latter 40 min of data were retained for analysis to minimize the effects from manipulating animals prior to recordings. Data from single animals were only retained in subsequent analyses if the following conditions were met: (1) the worm was visible in the video for at least half of the time the worm was recorded (cumulatively 30 min), (2) the worm was inside the bacterial lawn for at least one minute within the first 20 min of the video (before usable data was collected), (3) the plate was not bumped during the recording. All criteria were established before data collection.

## Quantitative locomotion analysis and HMM analyses

All quantitative analyses of locomotion and Hidden Markov Model building after behavioral tracking were performed in Python. For all analyses of animal locomotion and model-building, behavioral data from animals outside the food lawn was censored.

To classify roaming and dwelling states, speed and angular speed of animal centroid position was averaged into contiguous 10 s intervals. Roaming and dwelling states were classified as in *Flavell et al., 2013*. Briefly, two classes of intervals corresponding to high angular speed / low speed and low angular speed / high speed were identified and separated by a line drawn at y (mm/sec)=x (deg/sec) /450. Behavior can then be instantaneously classified into roaming intervals when values are above the line, or dwelling intervals, when values are below the line. A two-state categorical Hidden Markov Model was then trained on these behavioral sequences to generate roaming and dwelling hidden states using the SSM package (*Linderman et al., 2020*).

An Autoregressive Hidden Markov Model (AR-HMM) was trained to segment animal behavioral states on food using a different set of behavioral features relating to forward body movement and head movements: fraction of time moving forward per 10 s bin, midbody forward speed, midbody angular speed, head angular velocity relative to midbody and head radial velocity relative to midbody (*Figure 2*, *Figure 2—figure supplement 2*, see above for feature definitions).

Formally, at each time step $t$, we have discrete hidden states $z_t \in 1, 2, \ldots K$ that follow Markovian dynamics, $z_{t+1}|z_t, \{\pi_k\}_{k=1}^{K} \sim \pi_{z_t}$, where $\{\pi_k\}_{k=1}^{K}$ is the transition matrix and $\pi_k \in [0, 1]^K$ corresponds to the $k$-th row. Given hidden states $z_t$, the resulting feature dynamics are given by $x_t|x_{t-1}, z_t \sim N\left(A_{z_t}x_{t-1} + b_{z_t}, Q_{z_t}\right)$, where $A_k$ and $Q_k$ are real 5x5 linear dynamics and covariance matrices, respectively. $b_k \in \mathbb{R}^5$ is the bias. The linear dynamics matrix A specifies a continuous flow on the feature space. The bias term $b$ can drive the dynamics in a particular direction. In the case where A is all zeros, the system has no dynamics and this amounts to a Hidden Markov Model with Gaussian emissions. AR-HMMs were also trained using the SSM package (*Linderman et al., 2020*). AR-HMM performance was evaluated by calculating the ratio of the log-likelihood of held-out test data set using the AR-HMM same data under the AR-HMM and a multivariate Gaussian model $x_t \sim N\left(\mu, \Sigma\right)$.

## Sample size determination, replicates and group allocation

The number of animals in each experiment is detailed in the figure legends. Assays were typically repeated across at least 2 days of experiments to account for day-to-day variability. Control animals were always run on the same days as experimental animals.

Using G*POWER 3.1, we chose sample sizes based on the desired ability to detect an effect size of 1 with 80% power and a 5% alpha, yielding a minimum value of n=18 animals per group for a two-sample unmatched comparison of means and n=11 animals per group for a matched comparison of means (*Faul et al., 2009*).

## Quantification and statistical analysis

All comparisons in the fraction of time roaming were performed after logit-transformation $logit\left(p\right) = \ln\left(\frac{p}{1-p}\right)$, where p is the fraction of time spent roaming per animal.

Either Wilcoxon rank sum test or paired t-tests were used to compare matched data (roaming speed at –3 to –1 min vs. last 30 s before leaving, number of lawn leaving events during light OFF vs. ON, fraction of time roaming during light OFF vs. ON).

The Kolmogorov-Smirnov test was used for comparing cumulative distributions.

If multiple pairwise tests were done, multiple hypothesis correction was always performed.

Statistical details for each experiment are described in the figure legends. The p values resulting from all statistical tests performed in the paper can be found in *Supplementary file 4*.

## Acknowledgements

We thank Cheryl Mai for initial analysis of optogenetics experiments. We thank Philip Kidd, Audrey Harnagel, Yarden Wiesenfeld, and Steven Flavell for thoughtful discussions and comments on the manuscript. This work was supported by the Chan Zuckerberg Initiative and by an NIH F31 Predoctoral NRSA Fellowship to E.S.

## Additional information

### Funding

| Funder | Grant reference number | Author |
| --- | --- | --- |
| Chan Zuckerberg Initiative | | Cornelia I Bargmann |
| National Institute of Mental Health | 1F31MH113313 | Elias Scheer |

The funders had no role in study design, data collection and interpretation, or the decision to submit the work for publication.

### Author contributions

Elias Scheer, Conceptualization, Resources, Software, Investigation, Visualization, Methodology, Writing – original draft, Writing – review and editing; Cornelia I Bargmann, Conceptualization, Supervision, Funding acquisition, Methodology, Writing – review and editing

### Author ORCIDs

Elias Scheer ⓘ https://orcid.org/0000-0003-3462-3970
Cornelia I Bargmann ⓘ http://orcid.org/0000-0002-8484-0618

Reviewer #1 (Public Review): https://doi.org/10.7554/eLife.88657.3.sa1
Reviewer #2 (Public Review): https://doi.org/10.7554/eLife.88657.3.sa2
Reviewer #3 (Public Review): https://doi.org/10.7554/eLife.88657.3.sa3
Author Response https://doi.org/10.7554/eLife.88657.3.sa4

## Additional files

### Supplementary files

• Supplementary file 1. Sensory neurons and endocrine systems implicated in roaming and leaving behavior.

• Supplementary file 2. Lawn Leaving Events Per minute Roaming/Dwelling.

• Supplementary file 3. Strains and Plasmids.

• Supplementary file 4. p-values and Statistical Tests.

• MDAR checklist

### Data availability

All primary behavioral data and relevant code for data analysis are available at Dryad (https://doi.org/10.5061/dryad.47d7wm3jf) and Github (https://github.com/BargmannLab/Scheer_Bargmann2023; copy archived at *Scheer and Bargmann, 2023*) without restriction. Source data files contain the summarized data for all plots in Figures 1-6. All strains and plasmids used are found in Table 3: Strain details are available from the corresponding author without restrictions.

The following dataset was generated:

| Author(s) | Year | Dataset title | Dataset URL | Database and Identifier |
|---|---|---|---|---|
| Scheer E, Bargmann CI | 2023 | Sensory neurons couple arousal and foraging decisions in *C. elegans* | https://doi.org/10.5061/dryad.47d7wm3jf | Dryad Digital Repository, 10.5061/dryad.47d7wm3jf |

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
