## [Editor Report · eLife assessment]

This is an **important** study on how behavioral context affects decision making in the nematode *C. elegans*. Behavioral analyses at multiple time scales combined with genetic and neuronal manipulations revealed how arousal states affect decision making. The results and interpretations are **convincing**. This work will be of interest to both neuroscientists and ecologists.

---

## [Referee Report · Reviewer #1 (Public Review)]

Genetic, physiological, and environmental manipulations that increase roaming increase leaving rates. The connection between increased roaming and increased leaving is lost when tax4-expressing sensory neurons are inactivated. This study is conceptually important in its characterization of worm behaviors as time-series of discrete states, a promising framework for understanding behavioral decisions as algorithms that govern state transitions. This framework is well-established in other animals, but relatively new to worms.

A key discovery is that lawn leaving behavior is probabilistically favored in states of behavioral arousal. I like the use of response-triggered averages (triggered on leaving events) that illustrate a "state-dependent receptive field" of the behavioral response. Response-triggered averages are common in sensory neuroscience, used, for example, to characterize the diverse "stimulus-dependent receptive fields" of different retinal ganglion cell types. It's nice to adapt the idea to illustrate the state-dependence of behavioral state transitions.

The simplest metric of arousal state is crawling speed. When animals crawl faster, they are more likely to leave lawns. A more sophisticated metric of behavioral context is whether the animal is in a "roaming" or "dwelling" state, two-state HMM modeling from previous work (Flavell et al., 2013). Roaming animals are more likely to leave lawns than dwelling animals. Different autoregressive HMM tools can segment worm behavior into 4-states. Also with ARHMMs, the most aroused state is again the state that promotes lawn-leaving. HMM analysis disentangles effects that were lumped by the simpler metric of overall speed.

The authors use diverse environmental, genetic, and optogenetic perturbations to regulate the roaming state, thereby regulating the statistics of leaving in the expected manner. One surprise is that feeding inhibition evokes roaming and lawn-leaving in both pdfr-1 and tph-1 mutants, even though the tph-1-expressing NSM neurons have been shown to sense bacterial ingestion and food availability.

Another surprise is that evoking roaming does not evoke leaving in tax-4 mutants. Without sensory neuron activity, worms are only more likely to roam for a minute before leaving rather than roaming for several minutes before leaving like wild-type (Figure 6C). ASJ seems to be the most important sensory neuron in this coupling between roaming and leaving (which is uncoupled when sensory neurons are inactivated).

---

## [Referee Report · Reviewer #2 (Public Review)]

Here, the authors use quantitative behavioral analyses to describe in unprecedented detail the various behavioral choices animals make when encountering the lawn edge. They report that leaving the lawn is a rare outcome compared to other choices such as pausing or reversing back into the lawn. It occurs predominantly out of the roaming state and has a characteristic preceding fast crawling profile. They developed a refined analysis method, the result of which suggests that the arousal state of animals on food can be described by a 4-state behavior (as opposed to the 2-state roaming - dwelling classification); leaving the lawn occurs predominantly from "state 3", which corresponds to the highest level of arousal during roaming. They further show that various manipulations, such as optogenetic inhibition of feeding, stimulation of RIB neurons, or mutations of neuromodulator pathways, all of which have previously been reported to affect crawling speed and/or roaming/dwelling, maintain the coupling between roaming states and leaving, suggesting a dedicated mechanism for coupling leaving to the roaming state. Finally, they use genetics to implicate chemosensory neurons as neuronal circuit elements mediating this coupling.

How arousal states affect decision making is an active area of neuroscience research; therefore, the current manuscript will impact the field beyond the small community of *C. elegans* researchers. Also, in the past, roaming/dwelling and leaving have been treated as independent behaviors; the current manuscript is very intriguing, demonstrating both the interconnectedness of different behavioral programs and the importance of the animal's behavioral context for specific decisions.

In this current revision and, the authors have made a good effort at addressing most of my previous comments, especially to clarify the sample sizes and how independent assays were performed.

My major concern, however, remains: when leaving animals apparently accelerate their locomotion speed starting about 30s prior to the leaving events (Fig. 2A, D, G). By the authors' analysis, these episodes are assigned to roaming or 'state 3'. Note, that even within these states the behavior seems to be distinctively faster than baseline roaming- or 'state 3'- speed (Fig. 2A, D, G). If leaving is indeed preceded by a stereotypic acceleration phase, this phase should be assigned to the leaving event, not to roaming or 'state 3'. If this is done, the distribution of roaming dwelling states prior to acceleration-leaving could get closer to 50/50 (draw a vertical line at 30s onto Figure 2C, and then count the fraction of prior roaming-dwelling states). I would conclude that the probability of leaving is also high out of the dwelling-state. This interpretation challenges the major conclusion of the study, which is that the roaming behavioral state is a major determinant of the leaving decision. The analysis in Figure 2 S1E shows interesting results hinting that leaving is indeed not fully independent of the roaming history, but does not directly address the issue described above.

I think that the work is otherwise overall very well done and the results are extremely interesting. But I would interpret the results differently unless the authors provide a more tailored analysis that rules out my concern.

---

## [Referee Report · Reviewer #3 (Public Review)]

Scheer and Bargmann use a combination of computational and experimental approaches in *C. elegans* to investigate the neuronal mechanisms underlying the regulation of foraging decisions by the state of arousal. They showed that, in *C. elegans*, the decision to leave food substrates is linked to a high arousal state, roaming, and that an increase in speed at different timescales preceded the food leaving decisions. They found that mutants that exhibit increased roaming also leave food substrates more frequently and that both behaviors can be triggered if food intake is inhibited. They further identify a set of chemosensory neurons that express the transduction channel tax-4 that couple the roaming state and the food-leaving decisions. The authors postulate that these neurons integrate foraging decisions with behavioral states and internal feeding cues.

The strength of the paper relies on using quantitative and detailed behavioral analysis over multiple time scales in combination with manipulation of genes and neuron to tackle the state-dependent control of behavioral decisions in *C. elegans*. The evidence is convincing, the analysis rigorous, and the writing is clear and to the point.

---

## [Author Response]

In this paper, we examine the behavioral context that generates foraging decisions at the boundaries of food patches in the nematode *C. elegans*. By analyzing animal locomotion at high spatial and temporal resolution, we identify discrete behavioral responses to encountering the edge of a food patch that can be understood as a decision: either to remain inside the food patch or to leave it. We find that the decision to leave a food patch is associated with increased behavioral arousal that unfolds on long and short timescales. The coupling of increased arousal to lawn leaving decisions is preserved across genetic, neuronal, and environmental manipulations that alter global arousal levels. However, genetic inactivation of a set of chemosensory neurons disrupts the coupling of arousal and lawn leaving, revealing a potential site of integration between internal signals and external sensation that governs foraging.

We appreciate the reviewers’ thoughtful engagement with this work. In addition to modifications in the text to address minor concerns and ambiguities, we have conducted new analyses and made text and figure edits to strengthen or explain our conclusions. We have also investigated possible confounding explanations to our interpretation of the data.

In newly added analysis, we show that increased arousal does not result in increased proximity to the lawn boundary, which would be a trivial reason why roaming animals leave more than dwelling ones (new Figure 2-Supplement 4).

We also addressed the concern that classifying the brief speed acceleration motif as a roaming state would inflate the apparent coupling of roaming to leaving. By measuring the duration of roaming states prior to leaving, we in fact found the opposite: roaming states that precede leaving are slightly longer than other roaming states, not short acceleration events (new Figure 2-Supplement 1E).

The reviewers also asked reasonable questions about variability between batches of experiments. In particular, reviewers pointed out high levels of roaming in wild type controls accompanying npr-1 mutants. Indeed, the simultaneously-tested wild type animals roamed more than usual in this experiment (Fig. 4C,K) and less than usual in other panels (Fig. 4A,B,I,J) in these small datasets. There is more to do here, but the results support the general point that roaming and leaving are correlated in several neuromodulatory mutants that regulate roaming. We have included a new sentence in the Figure 4 legend to draw the reader’s attention to the potential limitations of these results, and to explicitly state that results should not be compared across panels. Similarly, there is more to be done to understand tax-4, as we did not test all tax-4-expressing sensory neurons for their effects on roaming and leaving.

In private comments, reviewers also asked about experimental design and statistics and were concerned that certain assays conducted on just a few days may not represent independent experiments. We have updated the Methods section to improve the description of the behavioral experiments, including more information about the behavioral chambers and imaging conditions. We note that for all experiments we tested all relevant genotypes in the same batches and days, enabling comparisons of experimental animals with matched controls conducted at the same time.

Reviewers asked us to compare our results to those generated by Rhoades, et al. (2019) and Cermak, et al. (2020). To the best of our knowledge, our results are fully consistent with those studies. The study by Rhoades and co-authors is primarily concerned with behavioral slowing upon first encountering a food patch, and thus does not include data regarding roaming or lawn leaving (Rhoades et al., 2019). As we mention in the text, we were initially surprised that tph-1 did not eliminate regulation of roaming by feeding, but there are straightforward explanations (redundant transmitters, other neurons). tph-1 did have a significant, albeit small, effect. The study by Cermak and co-authors presents an alternative Hidden Markov Model that uses whole animal postures to segment on-food behavior into 9 states including 8 dwelling states and a single roaming state (Cermak et al., 2020); we refer to this analysis in the discussion. Cermak’s paper and ours differ in experimental conditions, the behaviors measured, and the models used to analyze them. The animals in the Cermak paper are exposed to a large bacterial lawn of uniform density, whereas animals in our study are recorded on small bacterial lawns with thick edges. The analysis tools also differ in their use of animal posture (Cermak only) and autoregressive dynamics (our work only). Further studies of the neurons and molecules involved may help to fully harmonize these models.

References

Cermak, N., Yu, S.K., Clark, R., Huang, Y.C., Baskoylu, S.N., and Flavell, S.W. (2020). Whole-organism behavioral profiling reveals a role for dopamine in statedependent motor program coupling in *C. elegans*. Elife 9, 1–34.

Rhoades, J.L., Nelson, J.C., Nwabudike, I., Yu, S.K., McLachlan, I.G., Madan, G.K., Abebe, E., Powers, J.R., Colón-Ramos, D.A., and Flavell, S.W. (2019). ASICs Mediate Food Responses in an Enteric Serotonergic Neuron that Controls Foraging Behaviors. Cell 176, 85-97.e14.